# Risk assessment for airborne disease transmission by poly-pathogen aerosols

**Freja Nordsiek**[1], **Eberhard Bodenschatz**[1,2,3]*, **Gholamhossein Bagheri**[1]

**1** Max Planck Institute for Dynamics and Self-Organization (MPIDS), Göttingen, Niedersachsen, Germany,
**2** Institute for Dynamics of Complex Systems, University of Göttingen, Göttingen, Niedersachsen, Germany,
**3** Laboratory of Atomic and Solid State Physics and Sibley School of Mechanical and Aerospace Engineering,
Cornell University, Ithaca, New York, United States of America

* lfpn-office@ds.mpg.de

journal.pone.0248004

NETHERLANDS

**Data Availability Statement:** All relevant data are
within the manuscript and its Supporting
information files.

**Funding:** This work has been funded by the BMBF
as part of the B-FAST (Bundesweites Netzwerk

## Abstract

In the case of airborne diseases, pathogen copies are transmitted by droplets of respiratory
tract fluid that are exhaled by the infectious that stay suspended in the air for some time and,
after partial or full drying, inhaled as aerosols by the susceptible. The risk of infection in
indoor environments is typically modelled using the Wells-Riley model or a Wells-Riley-like
formulation, usually assuming the pathogen dose follows a Poisson distribution (mono-path-
ogen assumption). Aerosols that hold more than one pathogen copy, i.e. poly-pathogen
aerosols, break this assumption even if the aerosol dose itself follows a Poisson distribution.
For the largest aerosols where the number of pathogen in each aerosol can sometimes be
several hundred or several thousand, the effect is non-negligible, especially in diseases
where the risk of infection per pathogen is high. Here we report on a generalization of the
Wells-Riley model and dose-response models for poly-pathogen aerosols by separately
modeling each number of pathogen copies per aerosol, while the aerosol dose itself follows
a Poisson distribution. This results in a model for computational risk assessment suitable for
mono-/poly-pathogen aerosols. We show that the mono-pathogen assumption significantly
overestimates the risk of infection for high pathogen concentrations in the respiratory tract
fluid. The model also includes the aerosol removal due to filtering by the individuals which
becomes significant for poorly ventilated environments with a high density of individuals,
and systematically includes the effects of facemasks in the infectious aerosol source and
sink terms and dose calculations.

## Introduction

It is well known that some diseases such as influenza, the common cold, *Mycobacterium tuber-
culosis*, measles, and Severe Acute Respiratory Syndrome Coronavirus 1 (SARS-CoV-1) are
airborne; meaning they can be transmitted by particles (also called liquid droplets, aerosols, or,
if completely dried, droplet nuclei) exhaled by infected individuals that stay suspended in the
air for some time rather than immediately falling to the ground. These particles come from the
fluid of the lungs, vocal chords, mouth, and nose; which hereafter are all noted as "respiratory

Ange- wandte Surveillance und Teststrategie) project (01KX2021) within the NUM (Netzwerk Universitätsmedizin) and the Max Planck Society. The funders had no role in study design, data collection and analysis, decision to publish, or preparation of the manuscript.

**Competing interests:** The authors have declared that no competing interests exist.

tract". While these particles that stay airborne as well as larger ones that tend to fall on the ground and surfaces are all drops/droplets unless they have completely dried out to solid solute and they are technically aerosols (albeit, sometimes large), the literature usually refers to small airborne ones as aerosols and the larger ones that don't get suspended in the air as drops/droplets, which we shall do here as well. Note that these diseases can have additional transmission pathways, which can be more or less significant depending on the circumstances. Whether Severe Acute Respiratory Syndrome Coronavirus 2 (SARS-CoV-2) is an airborne disease or not and the relative importance of the airborne pathway to the pathway of exhaled droplets too large to stay airborne ballistically getting on susceptible individuals and surfaces have been topics of ongoing discussion and debate throughout the pandemic [1–3]. Due to the possibility that SARS-CoV-2 might be an airborne disease among other transmission pathways, the SARS-CoV-2 pandemic has brought an increased interest in airborne disease transmission dynamics and models.

The risk of getting infected from such airborne particles for an individual or a population has been the subject of numerous studies and analyzes [1, 4–10]. Many of the transmission mitigation strategies rely on results obtained by models that take into account a variety of factors to assess the likelihood of transmission, a good example of which is the World Health Organization's 2009 guidelines *Natural Ventilation for Infection Control in Health-Care Settings* [11]. Two well-known families of models are dose-response and Wells-Riley models, which have been extensively used to model the spread of airborne diseases [12].

There are several dose-response models for various diseases in existence which consider the risk of infection for an average dose of pathogen copies, taking full account of the counting statistics [13]. Two common models are the exponential and beta-Poisson models, which are described in great detail by Haas, Rose & Gerba [13]. Many diseases follow the exponential model, which has the added simplicity of having only a single adjustable parameter. Both the exponential and beta-Poisson models assume that the minimum number of pathogen copies required for infection, the threshold, is one; but other models exist for non-unity thresholds. Both models, along with many others, assume that the number of pathogen copies absorbed follows a Poisson distribution; though modification of the exponential model for doses following a beta or gamma distribution has been conducted [5].

The Wells-Riley model, in its original form, takes the steady state balance of sources and sinks of airborne infectious pathogen copies (in units of quanta) over a period of time in a well-mixed indoor environment such as a room or several rooms connected via ventilation (homogeneous concentration assumption) to calculate the average dose received by susceptible individuals over a time period, which is then run through an exponential dose-response model [4]. The original model measures pathogen copies in units of quanta, which is defined as $ID_{63.21}$ pathogen copies [12]. Sources such as exhalation by infectious individuals in the environment and air exchange with other environments with infectious aerosols and sinks due to fluxes with outside, filtering by the ventilation, filtering by masks, inactivation, settling, and deposition have all been considered as well as full temporal modelling of the infectious aerosol concentration rather than assuming steady-state [1, 4–10]. At the model's heart, it is essentially a conservation of infectious aerosols model, choosing some sources and sinks to explicitly include and considering others to be negligible, to get the pathogen concentration and then the average inhaled dose, before using a dose-response model (usually the exponen-tial model) for the infection risk. Note, in the literature the term "Wells-Riley model" is some-times used to refer only to when this formulation is used with an exponential model, and the terms "Wells-Riley equation" and "dose-response model" used if other dose-response models are used instead (e.g. [12]). We will use the term "Wells-Riley formulation" to refer to both.

Wells-Riley formulations are a statistical treatment of airborne disease transmission. Underneath its source, sink, and respiratory tract absorption parameters (as well as the choices of which to include and exclude) and its well-mixed assumption and their caveats/limitations are a mix of fluid dynamics with inertial particles (aerosols), the biological processes of the respiratory tract and diseases, thermodynamics, aerosol chemistry, human behavior and safety interventions (e.g. wearing masks), etc. This includes breathing rates for different activities [14–17]; the dynamics of exhaled puffs and the particles within them by breathing, speech, coughing, etc. [18–21]; the generation and ejection of aerosols and larger droplets by breathing, speech, coughing, etc. [16, 19, 22, 23]; aerosol/droplet growth/evaporation in response to temperature and humidity [19, 24–28]; the dynamics of inertial particles in turbulence; mixing and transport [18–21, 29, 30]; ventilation and convection in indoor environments [30]; etc. There have been a number of recent papers that each go into several of these topics written during the course of the ongoing SARS-CoV-2 pandemic [19, 28–31], which while focused on SARS-CoV-2 are also applicable to other airborne diseases. In this manuscript, we will mostly focus on a statistical treatment.

In the past, various generalizations and improvements have been applied to the Wells-Riley formulation for situations beyond its original design and to address its limitations [12]. For example; Nicas, Nazaroff & Hubbard [9] included sink terms for pathogen inactivation, aerosol settling, and deposition as well as less than unitary efficiency of the respiratory tract absorbing infectious aerosols. Wells-Riley formulations have also been combined with SIR (Susceptible-Infectious-Removed) and SEIR (Susceptible-Exposed-Infectious-Removed) models [6, 32]. Noakes & Sleigh [33] made a stochastic model with compartmentalization of the environment into well-mixed subregions that have less mixing with other regions that can work for periods of time longer than the incubation period. Recent Wells-Riley based analyzes during the ongoing SARS-CoV-2 pandemic also include the effects of masks (such as [10]) unless they are investigating scenarios in which individuals are not wearing any mask [1], though including the effect of masks predates the pandemic by decades [5–8].

One of the biggest assumptions of the Wells-Riley formulation is that the indoor environment is sufficiently well-mixed [1, 4, 7–10, 12, 33]. Essentially, it assumes that the infectious aerosol concentration is homogeneous enough that the concentration inhaled by susceptible individuals and at all sinks is approximately equal to the volume average concentration [1, 4, 7–10, 12, 33]. In reality, there can be concentration gradients on both large and small length scales in the environment. For example, the infectious aerosol concentration at close range directly in front of an infectious individual will usually be larger than the volume average of the whole environment since exhaled puffs from the infectious individual will not have dispersed much before inhalation. This means that a susceptible individual located where they can inhale such puffs would be at greater risk of infection than if they were not directly in the exhaled puffs of any infectious individuals. The nature of the ventilation plays a significant role in the validity of the assumption [30] The practice of social distancing, using fans to better mix the room, etc. all improve the quality of this assumption, but room conditions and people's proximity to each other in real-world situations can be far away from the well-mixed state with everyone inhaling well-mixed air. We will make this same exact assumption in the model presented in this manuscript, and will neither be using nor developing corrections for close proximity between individuals and localized sinks and other sources, though the nature of partial corrections will be briefly discussed.

Besides the well-mixed assumption, there are several other assumptions associated with Wells-Riley formulations, which are not necessarily always true. As an example, there is an additional loss term that has not been fully considered yet that is the loss of the infectious aerosols absorbed by the individuals themselves, though the self-proximity depletion of infectious

aerosols in the vicinity of susceptible people has previously been mentioned as an effect to consider [29]. This is despite the fact that this is exactly the reason that susceptible individuals get infected. In some cases this can be safely neglected, e.g. if the combined breathing volume exchange rate of all individuals in the environment is negligible compared to that of ventilation. But in a poorly ventilated room with many individuals inside, this sink term must be taken into account—not incorporating it leads to false risk predictions.

Another large assumption is that the absorbed doses follow a Poisson distribution, which is implicit in the use of the exponential dose-response model even if not stated explicitly [1, 4, 6, 9, 10], though there has been work on doses following beta and gamma distributions [5]. The Poisson distribution assumption requires that the pathogen-carrying aerosols have at most one pathogen inside, i.e. a mono-pathogen assumption. However, this assumption is violated if the pathogen concentration in an infectious individual's respiratory tract is high. For this poly-pathogen situation the Wells-Riley formulation and the dose-response models must be generalized to consider a larger number of pathogen in an individual aerosol explicitly. We will use the term multiplicity to refer to the number of pathogen copies in an aerosol.

Ignoring multiplicity causes the infection risk to be overestimated even though the expected average pathogen dose does not change. Using a modified version of the worked example later in this manuscript, Fig 1 shows this effect on the time required to reach a 50% infection risk for different pathogen concentrations in the respiratory tract fluid with and without considering multiplicity. For low pathogen concentrations and small infection probabilities per pathogen, ignoring multiplicity has only a small effect. But for high pathogen concentrations and/or pathogen copies with a high infection probability per pathogen, ignoring multiplicity has a significant impact. For a respiratory tract pathogen concentration of $10^{11}$ cm$^{-3}$ where the average number of pathogen copies per aerosol is approximately 6500 for a 50 μm in diameter at production, if the single pathogen infection probability ($r$) is large enough that multiplicity matters, this means taking into account multiplicities up to approximately 7000.

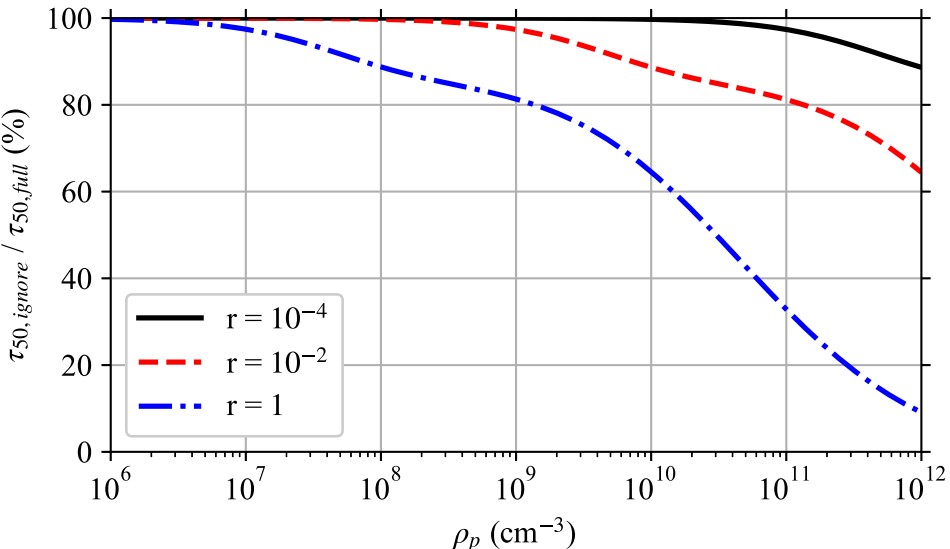

**Fig 1. Effect of ignoring multiplicity.** Ratio of the time required to reach a 50% infection risk when multiplicity is ignored $\tau_{50,ignore}$ to when it is fully accounted for $\tau_{50,full}$ for single pathogen infection probabilities $r$ (an average dose of $r^{-1}$ Poisson distributed pathogen copies gives a mean infection risk of 63.21%) and different pathogen concentrations $\rho_p$ in the respiratory tract fluid of the infectious individual as in the worked example later in the manuscript with a disease following the exponential model, but at steady-state with just the speaking mask-less infectious individual and the risk to a mask-less susceptible individual whose exposure starts after steady state is reached. This is a simplified version of Fig 5.

In this manuscript, we will consider the following generalizations and modifications to the Wells-Riley formulation:

- Fully accounting for the multiplicity of pathogen copies in aerosols and the effect on the dose-response models.

- Additional sink terms due to the filtering of air by people inhaling and then exhaling it back out, including the effects of masks.

- Working exclusively in units of pathogen copies and aerosols instead of quanta (note, quantum is undefinable when accounting for multiplicity).

We will first generalize dose-response models that assume Poisson distributed doses for the distribution that results from poly-pathogen aerosols being present. Then we will develop the general pathogen concentration model that is a generalization of the Wells-Riley formulation. This results in a linear inhomogeneous coupled system of ODEs (Ordinary Differential Equations) for each initial aerosol diameter at production (diameter when exhaled), with one equation for each multiplicity that must be considered. We then derive the general solution, and then simplify the general solution for coefficients that are constant in time. Requirements and heuristics are developed for finding the appropriate cutoff in the multiplicity, $M_c$. This is important because the number of ODEs to solve is equal to $M_c$; and the computational effort scales as $\mathcal{O}(M_c^2)$ for the numerical solution, or worse than for $\mathcal{O}(M_c^2)$ or $\mathcal{O}(M_c^3)$ for the different analytical solutions for coefficients constant in time. Some circumstances allow small $M_c = 1$ or close to one. We consider a full hypothetical example situation for SARS-CoV-2 with very high viral loads to apply the generalized Wells-Riley formulation developed in this manuscript. Finally, we discuss the effects of poly-pathogen aerosols, the filtering by the people in the environment, the effects of face-masks, and the model limitations and possible corrections. As a tool to aid solving the model presented in this paper, we wrote the PMADRA (Poly-Multiplicity Airborne Disease Risk Assessment) software suite (https://gitlab.gwdg.de/mpids-lfpn-public/pmadra).

## Fundamentals

Throughout this manuscript, we will use the Poisson distribution, which describes the probability of counting some number, $m$, of independent events/objects/etc. as a function of the ensemble mean of the number counted, $\mu$. The Probability Distribution Function (PDF) of the Poisson distribution is

$$P_P(\mu, m) = e^{-\mu}\left(\frac{\mu^m}{m!}\right) . \tag{1}$$

Most dose-response models assume that the number of pathogen copies absorbed follows a Poisson distribution. For the case of a dose-response model, the average number of pathogen copies absorbed over some period of time would be the $\mu$ and then $P_P$ would give the probability that a person absorbed exactly $m$ pathogen copies. For clarity in the rest of this manuscript, we will now define $\Delta$ to be the number of pathogen copies absorbed (instead of $m$) and the average number of pathogen copies absorbed is $\langle \Delta \rangle$, where we have used $\langle \cdot \rangle$ to denote the average. The use of a Poisson distribution for the doses requires that the pathogen copies are independent (i.e. no clumping); and as we will later show, that the number of pathogen copies in aerosols is assumed to be one or zero, which is generally assumed by existing models but won't be in the model presented in this manuscript.

Let $R(\Delta)$ denote the infection probability when exactly $\Delta$ pathogen copies are absorbed, and $\mathfrak{R}(\langle\Delta\rangle)$ denote the average infection probability when the average number of pathogen copies absorbed is $\langle\Delta\rangle$. For a disease where the threshold (minimum number of pathogen copies required for infection) is greater than one, the threshold must be included into the definition of $R(\Delta)$ such that it is zero for $\Delta$ less than the threshold, which makes $R(\Delta)$ be a piece-wise function.

There are two ways to construct $\mathfrak{R}(\langle\Delta\rangle)$ from $R(\Delta)$. We use the method of taking the sum over all possible $\Delta \in [1, \infty)$ of the product of the probability of absorbing each particular $\Delta$ and the resulting infection risk $R(\Delta)$ [12]. If the number of pathogen copies absorbed follows a Poisson distribution, then

$$\mathfrak{R}(\langle\Delta\rangle) = \sum_{\Delta=1}^{\infty} P_P(\langle\Delta\rangle, \Delta) R(\Delta) \,. \tag{2}$$

The other method instead considers the number of pathogen copies that survive to try to infect, $\Delta_i$, and does a double sum over $\Delta_i$ (starting from the threshold) and $\Delta$ of the product of the probability of the dose $\Delta$ and the probability of exactly $\Delta_i$ out of $\Delta$ surviving to try to infect [13] (this is **NOT** $R(\Delta)$). The two methods are equivalent, with this extra sum being implicitly included in the definition of $R(\Delta)$. This is why $R(\Delta)$ is a piece-wise function when the threshold is not one. For some models it may be easier to do this other method explicitly rather than try to construct $R(\Delta)$.

The exponential model assumes that all pathogen copies are identical, all people are equally vulnerable to infection, the pathogen copies are acting independently of each other, and that each pathogen has an equal probability of causing infection $r$ [13]. These assumptions implicitly mean that the threshold is one. Each pathogen has a probability $1 - r$ to not infect. Then the exponential model's infection risk for an exact dose $\Delta$ is just one minus the probability that all $\Delta$ pathogen copies did not infect.

$$R_E(\Delta) = 1 - (1 - r)^{\Delta} \,. \tag{3}$$

If the dose follows a Poisson distribution, then Eq (2) can be calculated for the exponential model [12], yielding

$$\mathfrak{R}_E(\langle\Delta\rangle) = 1 - e^{-r\langle\Delta\rangle} \,. \tag{4}$$

Note that often, the parameter $D \equiv 1/r$ is used instead of $r$ (the symbol $k$ is also used [34]), which is the $ID_{63.21}$ (Infective Dose required for 63.21% chance of infection). We will be making non-Poissonity corrections to this later.

The beta-Poisson model is essentially the exponential model but instead of considering everyone to be equally vulnerable, each person has their own value for $r$ which comes from the beta distribution [12, 13]. The beta distribution PDF [13] is

$$P_B(r) = \frac{\Gamma(\varepsilon + \theta)}{\Gamma(\varepsilon)\Gamma(\theta)} r^{\varepsilon-1}(1 - r)^{\theta-1} \,, \tag{5}$$

where $r \in [0, 1]$ and the symbols $\varepsilon$ and $\theta$ have been used in place of the conventional alpha and beta parameters respectively to avoid clashing with symbols used later in this manuscript. This means that to get the mean infection risk for a beta-Poisson model $R_{BP}(\Delta)$, we must include an integral over all $r \in [0, 1]$. Specifically,

$$R_{BP}(\Delta) = \int_0^1 P_B(r) R_E(\Delta) dr \,. \tag{6}$$

Since the integral commutes with the sums used to calculate $\mathfrak{R}(\langle\Delta\rangle)$, the integral can be calculated as an outer integral rather than an inner integral yielding [13]

$$\mathfrak{R}_{BP}(\langle\Delta\rangle) = \int_0^1 P_B(r)\mathfrak{R}_E(\langle\Delta\rangle)dr \, . \tag{7}$$

Wells-Riley formulations, both the original model and many subsequent uses, measure pathogen copies in units of quanta [1, 4, 8–10, 12, 33]. A quanta is defined as $ID_{63.21}$ pathogen copies [12]. This means that one quantum is equal to $D = 1/r$ pathogen copies. For the case of $r = 1$ such as *Mycobacterium tuberculosis*, one quantum is one pathogen [9, 12]. Using these units, the exponential model from Eq (4) becomes

$$\mathfrak{R}_E(\langle\mathbb{Q}\rangle) = 1 - e^{-\langle\mathbb{Q}\rangle} \, , \tag{8}$$

where $\mathbb{Q}$ is the number of absorbed quanta [1, 4, 8–10, 12, 33].

Let $N_I$ be the number of infectious individuals, $\sigma$ be the average production rate of infectious quanta per infectious individual, $\lambda$ be the volumetric breathing rate of susceptible individuals, $Q$ be the volumetric rate that clean air is brought into the particular indoor environment, and $\tau$ be the time period of exposure of susceptible individuals. Then, in its simplest form, the Wells-Riley Model's infection probability for time periods smaller than the incubation period of the disease [4] is

$$\mathfrak{R}_{WR}(\tau) = 1 - \exp\left[-\left(\frac{N_I\sigma}{Q}\right)\lambda\tau\right] \, . \tag{9}$$

For time periods longer than the incubation period of the disease, one must either break the time period into subintervals smaller than the incubation period [4] or model both $\mathfrak{R}_{WR}$ and the number of infectious and susceptible individuals over time with a SIR or SEIR model [6, 32].

## Dose-response models for poly-pathogen aerosols

### General

If the pathogen concentration in an infectious individual's respiratory tract fluid $\rho_p$ is low enough, almost all exhaled pathogen copies will be the only pathogen in their aerosols, i.e. mono-multiplicity aerosols, and poly-multiplicity aerosols can reliably be ignored. We will use the tailing subscript $k$ to denote aerosols with $k$ pathogen copies inside them. An aerosol cannot contain more pathogen copies than will fit in its volume, and there is a limit to how large an aerosol/droplet a person can exhale. Let $M$ be the maximum number of pathogen copies that can fit in the largest aerosol/droplet that can possibly be exhaled. This is the hard cutoff/limit on $k$. There also exists a soft cutoff/limit $M_c \le M$ for which contributions of aerosols with $k > M_c$ is negligible. In a worst case $M_c = M$, but in practice it is much lower since the pathogen volume fraction of respiratory tract fluid is quite low even at the upper pathogen load for some diseases and the largest droplets don't stay airborne and ballistically fall to the ground. For example, SARS-CoV-2 at the very upper end of its concentration range at $10^{11}$ cm$^{-3}$ [35, 36] would give a volume fraction of approximately $5 \times 10^{-5}$, if we treat the virus as a 100 nm sphere (approximate size of the SARS-CoV-2 virus [37]). This is important because an aerosol with a diameter of 1 μm could contain up to approximately 740 spherical pathogen copies with diameter 100 nm, if we assume hard-sphere packing (packing fraction of 74%). An aerosol with a

diameter of 10 μm could contain up to approximately $7.4 \times 10^5$ of the same pathogen copies for the same packing fraction.

To properly account for higher multiplicities, we must consider the separate doses for each multiplicity. Let $\Delta_k$ be the number of **pathogen copies** absorbed from aerosols with multiplicity $k$, and let $m_k$ be the number of **aerosols** absorbed with multiplicity $k$. The aerosol and pathogen doses are related by $\Delta_k = k m_k$. The total pathogen dose from all aerosols is just the sum of the doses for each multiplicity, which is $\Delta = \sum_{k=1}^{\infty} \Delta_k$. Let $\mu_k = \langle m_k \rangle = \langle \Delta_k \rangle / k$ be the average number of absorbed aerosols with multiplicity $k$.

As long as the aerosols are randomly distributed in space (well-mixed with no clustering nor avoidance), then the PDF of each $m_k$ follows a Poisson distribution with mean $\mu_k$. Since $\Delta_k = k m_k$, the PDF of $\Delta_k$ is not a Poisson distribution for $k > 1$. It is instead a scaled-Poisson distribution of the form

$$P_k(\mu_k, \Delta_k) = \begin{cases} P_P\left(\mu_k, \frac{\Delta_k}{k}\right) & \text{if } \Delta_k \mod k = 0 \,, \\ 0 & \text{otherwise} \,. \end{cases}$$

The deviation from the Poisson distribution is most visible in the fact that this distribution has holes. For example with $k = 2$, $P_k = 0$ for all odd $\Delta_k$. Since $\Delta$ is the sum of a Poisson distribution for $k = 1$ and some number of possibly non-negligible scaled-Poisson distributions, the PDF of $\Delta$ will not be a Poisson distribution unless the contributions from $k > 1$ are negligible compared to $k = 1$. So we can't just naively put the expected average dose into dose-response models expecting a Poisson distribution.

Instead, we must change the summation in Eq (2) to get the infection risk $\mathfrak{R}$. Let us consider the $p$'th moment, $\mathcal{M}_p$, of the infection probabilities as a function of the average aerosol doses $\mu_k$ (note, we use $p$ in later sections of this manuscript as a summation index). To determine $\mathcal{M}_p$, we must sum over all possible combinations of exact aerosol doses $m_k$ of each multiplicity for $k \in [1, \infty)$ of the product of the Poisson probabilities of each $m_k$ and the infection risk for the dose raised to the power of $p$. This is

$$\mathcal{M}_p(\mu_1, \cdots, \mu_\infty) = \overbrace{\sum_{m_1=0}^{\infty} \cdots \sum_{m_\infty=0}^{\infty}}^{\text{all combinations}} \overbrace{\left[ \prod_{k=1}^{\infty} P_P(\mu_k, m_k) \right]}^{\text{probability of dose}} \left[ R \Big( \underbrace{\overbrace{\sum_{k=1}^{\infty} k m_k}^{\text{infection probability}}}_{\text{pathogen dose}} \Big) \right]^p, \tag{10}$$

where we have written out the dose $\Delta$ inside $R$. The mean infection risk is the first moment ($p = 1$), which is

$$\mathfrak{R}(\mu_1, \cdots, \mu_\infty) = \sum_{m_1=0}^{\infty} \cdots \sum_{m_\infty=0}^{\infty} \left[ \prod_{k=1}^{\infty} P_P(\mu_k, m_k) \right] R \left( \sum_{k=1}^{\infty} k m_k \right). \tag{11}$$

### Exponential model corrections

Then, putting $R_E$ from Eq (3) into Eq (11), the exponential model mean infection risk is

$$
\begin{aligned}
\mathfrak{R}_E(\mu_1, \cdots, \mu_\infty) &= \sum_{m_1=0}^{\infty} \cdots \sum_{m_\infty=0}^{\infty} \left[ \prod_{k=1}^{\infty} P_P(\mu_k, m_k) \right] \left[ 1 - (1-r)^{\sum_{k=1}^{\infty} k m_k} \right] \\
&= 1 - \sum_{m_1=0}^{\infty} \cdots \sum_{m_\infty=0}^{\infty} \prod_{k=1}^{\infty} e^{-\mu_k} e^{(1-r)^k \mu_k} e^{-(1-r)^k \mu_k} \frac{[(1-r)^k \mu_k]^{m_k}}{m_k!} \\
&= 1 - \exp\left[ -\sum_{k=1}^{\infty} (1 - (1-r)^k) \mu_k \right],
\end{aligned}
\tag{12}
$$

where the fact that the sum of all probabilities over the Poisson distribution is equal to one has been used extensively. The final sum has a finite number of terms due to the cutoff $M$ as long as the $\mu_k$ are finite for $k \leq M$. For small $M_c$, we can truncate the risk probability and get an easier to calculate approximation. Except for $M_c = 1$, this is different from Eq (4) due to the non-Poissonity in $\Delta$. The expression for the first few values of $M_c$ are

$$
\mathfrak{R}_E \approx \begin{cases}
1 - e^{-r\mu_1} & \text{if } M_c = 1, \\
1 - e^{-r\mu_1} e^{-r(2-r)\mu_2} & \text{if } M_c = 2, \\
1 - e^{-r\mu_1} e^{-r(2-r)\mu_2} e^{-r(3-3r+r^2)\mu_3} & \text{if } M_c = 3.
\end{cases}
\tag{13}
$$

### Beta-poisson model corrections

The integral over $r$ commutes with the sums in Eq (10). So as was with the case when multiplicity is not considered in Eq (7), we can get the moments by taking the result for the exponential model and integrating it times the beta distribution PDF over $r$. This is

$$
\mathcal{M}_{BP,p}(\mu_1, \cdots, \mu_\infty) = \int_0^1 P_B(r) \mathcal{M}_{E,p}(\mu_1, \cdots, \mu_\infty) dr.
\tag{14}
$$

Unfortunately, as is the case for when the dose is Poisson distributed [13], the integral cannot be solved analytically and must be solved numerically or approximated, though now it is harder with the extra terms for $M_c > 1$.

## General pathogen concentration model

### Looking ahead

Now that we have dose-response models corrected for the multiplicity via Eq (11), we must determine the average **aerosol** doses $\mu_k$ for each multiplicity before the infection risk can be calculated. We now generalize the Wells-Riley formulation for multi-pathogen aerosols to get this. In the following sections, we will describe the environment, people, aerosols, sources, sinks, etc. to get the model equations. Let $n_k(d_0, t)$ be the concentration density of aerosols with original diameter $d_0$ (diameter at production) and $k$ pathogen copies in them over time, which has units of $[L]^{-4}$ where $[L]$ is the unit of length since $n_k(d_0, t)dd_0$ is the concentration of infectious aerosols with original diameters between $d_0$ and $d_0+ dd_0$. To get a concentration, $n_k(d_0, t)$ must be integrated with respect to $d_0$.

In the end, we will get the following system of ODEs (Ordinary Differential Equations) in time $t$ and the original diameter at production $d_0$ for the $n_k$, which is

$$\frac{dn_k}{dt} = \overbrace{-\alpha(d_0, t)n_k}^{\text{sinks}} + \overbrace{(k+1)\gamma(t)n_{k+1} - k\gamma(t)n_k}^{\text{flux from inactivation}} + \overbrace{\beta_k(d_0, t)}^{\text{sources}} , \tag{15}$$

where $\alpha(d_0, t)$ is the sum of all sink term coefficients, $\beta_k(d_0, t)$ is the sum of all sources for each $k$, $\gamma(t)$ is the pathogen inactivation rate, and we have assumed that the time period considered is shorter than the incubation time of the disease. Then the combined source and sink terms are

$$\beta_k(d_0, t) = \beta_{r,k} + \beta_{I,k} , \tag{16}$$

$$\alpha(d_0, t) = \alpha_o + \alpha_r + \alpha_v + \alpha_g + \alpha_d + \alpha_{I,f} + \alpha_{S,f} + \alpha_{O,f} , \tag{17}$$

which don't depend on $n_k(d_0, t)$ (i.e. no quadratic or higher order terms), though they may depend on $t$. The different sources and sinks are summarized in Table 1. See their relevant sections for the meanings of their terms, their assumptions, and where they come from.

## Environment

Like most Wells-Riley formulations, we consider the infection risk in one sufficiently well-mixed indoor environment such as a room or set of rooms sufficiently coupled together with respect to their air that they have the same infectious aerosol concentration densities. And we assume that sources, sinks, and individuals are far enough apart from each other that the local concentration densities at their locations are approximately equal to the average concentration density in the whole environment. Note that the particular kind of ventilation has an impact on the validity of this assumption [30]. See the Discussion for when this assumption is not valid. The environment could also be split into coupled well-mixed zones with weaker mixing between them [7, 33], but that shall not be considered here.

Let the volume of the environment be $V$. Air is exchanged with outside, with other rooms, and circulated internally through the ventilation system. Let $Q_o$, $Q_r$, and $Q_v$ be the volumetric

**Table 1. Source and sink term summary.** Summary of all the source (the $\beta$) and sink (the $\alpha$) terms considered in this manuscript. "Individuals" is abbreviated as "ind." See their relevant sections for details on where they come from and the meanings of their terms.

| Term | Meaning | Form |
|------|---------|------|
| $\beta_{r,k}(d_0, t)$ | transport from other rooms | $q_r(t)n_{r,k}(d_0, t)$ |
| $\beta_{I,k}(d_0, t)$ | production by infectious individuals | $\frac{N_I}{V}\langle \lambda_I(t)n_{I,k}(d_0, t)[1 - E_{I,m,out}(d_0)]\rangle_I$ |
| $\alpha_o(t)$ | air exchange with outside | $q_o(t)$ |
| $\alpha_r(t)$ | air exchange with other rooms | $q_r(t)$ |
| $\alpha_v(d_0, t)$ | filtering by ventilation | $q_v(t)E_v(w(d_0, t)d_0)$ |
| $\alpha_g(d_0, t)$ | gravitational settling | $\approx \frac{1}{h}u_g(w(d_0, t)d_0)$ |
| $\alpha_d(d_0, t)$ | deposition on surfaces | found elsewhere |
| $\alpha_{I,f}(d_0, t)$ | filtering by infectious ind. inhaling | $\frac{1}{V}\sum_{j=1}^{N_I}\lambda_{I,j}(t)\left[1 - S_{I,m,in,j}(d_0, t)S_{I,r,j,k}(d_0, w, \lambda_{I,j})S_{I,m,out,j,k}(d_0)\right]$ |
| $\alpha_{S,f}(d_0, t)$ | filtering by susceptible ind. inhaling | $\frac{1}{V}\sum_{j=1}^{N_S}\lambda_{S,j}(t)\left[1 - S_{S,m,in,j}(d_0, t)S_{S,r,j,k}(d_0, w, \lambda_{S,j})S_{S,m,out,j,k}(d_0)\right]$ |
| $\alpha_{O,f}(d_0, t)$ | filtering by other ind. inhaling | $\frac{1}{V}\sum_{j=1}^{N_O}\lambda_{O,j}(t)\left[1 - S_{O,m,in,j}(d_0, t)S_{O,r,j,k}(d_0, w, \lambda_{O,j})S_{O,m,out,j,k}(d_0)\right]$ |

rate of air exchange with outdoors, other rooms, and the circulating ventilation of the environ-ment (ventilation system that pulls air out of the environment and puts it back in). These will be normalized by the environment volume; yielding $q_o \equiv Q_o/V$, $q_r \equiv Q_r/V$, and $q_v \equiv Q_v/V$ since target values of these parameters are often the design goals for HVAC systems.

## Aerosols

Consider the concentration of infectious aerosols over time. To be completely accurate, we need to consider the concentration density for each multiplicity $k$ as a function of time, current diameter $d$ while in the environment, and the solute content (including inactivated pathogen copies). We have to consider both $d$ and the solute content because an exhaled aerosol's equi-librium diameter is a function of its solute content, the humidity, and the temperature [27]. Higher solute concentrations decrease the vapor pressure of the aerosol, which allows equilib-rium to be reached as long as the environment isn't super-saturated or too close to saturated [26, 27]. For higher humidities, an aerosol will continue to grow by condensation indefinitely, though the growth rate slows towards a crawl for $d > 20$ μm [26, 38]. But such super-saturated conditions can cause clouds/fog, which rarely occur in indoor environments. So we will assume the environment is sub-saturated. If the environment is dry, the aerosols can evaporate at most to the point where they are purely precipitated solid with no water left. Note that as a drop (whether large or a small aerosol) dries, the solute fraction increases, until at some point the solute makes the shape non-spherical (not enough water to spherically encapsulate the insoluable components, solute causing anisotropy and/or inhomogeneity in the surface ten-sion, etc.). This will occur at a humidity no lower than the efflorescence relative humidity of the solute mix, where the soluble solutes will homogeneously nucleate and the water completely evaporates away. Infectious aerosols always have at least two components of the solute (whatever is in the respiratory tract fluid plus the pathogen/s), so there is the possibility of heterogeneous nucleation causing the water to completely evaporate away at a higher humidity.

This means that we have four different diameters to consider, which are

d.  current diameter in the environment (spherical equivalent diameter if it is completely dry or almost dry and the solute causes a non-spherical shape)

$d_e$.  equilibrium diameter in the environment

$d_0$.  wet diameter at production (original diameter), which determines the distribution of ini-tial multiplicities

$d_D$.  spherical equivalent dry diameter when all water is evaporated away and just solute remains (note that the aerosol may no longer be spherical, so the spherical equivalent diameter for the same volume must be used)

For any aerosol; $d_0$ and $d_D$ are fixed and never change as long as collisional-coalescence and shattering don't occur (can be treated as fixed if these processes are negligible), $d_e$ is dynamic in time if the environment's temperature and/or humidity changes, and $d$ is dynamic in time unless the environment's temperature and humidity exactly match those inside the respiratory tract at the point of production.

Small wet/nucleated aerosols respond very quickly to the humidity and temperature, evapo-rating/condensing to their equilibrium diameter in a very short period of time due to their high surface area to volume ratio [9, 25, 26]. Assuming the environment is well-mixed enough that the time between exhalation from an infectious individual and inhalation by any person is long compared to the evaporation/condensing time scale, we can make the approximation that

all aerosols are at their equilibrium diameter when in the environment ($d \approx d_e$). This means that when $d_e$ increases from $d_e = d_N$ (completely evaporated) to $d_e > d_N$ (wet/nucleated), we are assuming that the time the aerosols require to nucleate and grow to $d_e$ is short compared to other time scales in the model and therefore also make the approximation $d \approx d_e$ even when $d_e$ increases from $d_e = d_N$ to $d_e > d_N$. This means that we just need to worry about the equilibrium diameter and its changes, and not the non-immediate response to shifting equilibrium diameters. There is one complication, however. Aerosols will initially stay in the exhaled plume where the humidity is higher, so they won't reach the well-mixed equilibrium diameter till they leave the plume or the plume is diluted and mixed with the environment, which brings us back to the well-mixed environment assumption.

We will also make the assumption that the temperatures and humidities in different individuals' respiratory tracts (and the volume under their facemasks if they are wearing any) are similar enough and change negligibly enough over time that the equilibrium diameter in people's respiratory tracts is $d_0$. If the aerosols have not completely dried out in the environment ($d_e > d_D$), the aerosols will start to grow inside people's respiratory tracts back towards $d_0$ and thus $d \approx d_0$ inside the respiratory tract. But the time scale of breathing is short and for completely dried out aerosols it takes time to nucleate and grow back to $d_0$, which means that some fraction of dry aerosols might not reach $d = d_0$ while in the respiratory tract. However, we will make the assumption/approximation that dry aerosols have returned to their original diameter by the time they are exhaled back out if they were not absorbed in respiratory tract. This last approximation only affects the sink from individuals inhaling aerosols $\alpha_{C_f}(d_0, t)$ from Eq (42) if they are wearing masks, which is usually small compared to other sinks. When the individuals in the environment are wearing masks and the $\alpha_{C_f}(d_0, t)$ sinks dominate, then a better approximation or an explicit treatment of the diameter when exhaled should be used. Combined, our assumption/approximation is

$$d(t) \approx \begin{cases} d_e(t) & \text{if in the environment outside of the respiratory tract ,} \\ d_0 & \text{at re} - \text{exalation after inhalation .} \end{cases} \quad (18)$$

Let us define ratios between the remaining diameters: the evaporation ratio $w$, the dilution ratio $\delta$, and the initial solute ratio $\zeta$ as

$$w \equiv \frac{d_e}{d_0} , \quad (19)$$

$$\delta \equiv \frac{d_e}{d_D} , \quad (20)$$

$$\zeta \equiv \frac{d_D}{d_0} . \quad (21)$$

Note that $w$ and $\delta$ are potentially functions of time, as well as diameter due to the effect of surface curvature (through surface tension) on equilibrium vapor pressure [26, 27]. Also, different solutes have different molar densities, different practical osmotic coefficients, and maximum concentrations before they precipitate; and therefore different functional relationships between the saturation vapor pressure and the concentration [27]. So different solute compositions will cause $w$ and $\delta$ to be different even for aerosols with the same $\zeta$.

But, we will make the assumption that the value of $\zeta$ and the solute composition (except for the pathogen copies) is approximately constant from each infectious individual to the next and

over time with each infectious individual, and we will ignore the contribution of the pathogen copies (both active and inactivated) to the equilibrium vapor pressure and therefore $d_e$. We will also assume that $\zeta$ has no diameter dependence (i.e. attraction and repulsion of solutes from the liquid surface at production has a negligible effect on solute fraction and composition). With these approximations, we have a single constant value of $\zeta$ and single functions for $w$ and $\delta$, possibly over time and $d_0$ (or equivalently $d_D$), for all infectious aerosols in the environment.

This means we can choose to track one of $d_e$, $d_0$, or $d_D$ and always know the other two through the ratios that are the same for all infectious aerosols at the same moment of time with the same value of the chosen diameter parameter. Thus we have two independent variables, $t$ and one diameter parameter.

Processes such as gravitational settling, deposition, filtering or exchange by the ventilation, filtering by facemasks when inhaling are all functions of the current diameter, which is approximately $d_e$, making $d_e$ convenient. Additionally, any non-drying aerosol instrument can be used in the environment to measure $d_e$. But, because $d_e$ can change over time for a fixed $d_D$ or $d_0$, the equations for the aerosol concentration density in terms of $t$ and $d_e$ have a flux term (from evaporation/growth) with a partial derivative with respect to $d_e$; making the equations PDEs (Partial Differential Equations) which adds complications in the analysis. This can be seen by considering the total time derivative of the aerosol concentration density $\tilde{n}$ expressed in terms of $t$ and $d_e$, which is

$$\frac{d\tilde{n}(d_e, t)}{dt} = \frac{\partial \tilde{n}}{\partial t} + \frac{\partial \tilde{n}}{\partial d_e} \frac{d_e}{dt} . \tag{22}$$

Since $d_D$ and $d_0$ are fixed for a given aerosol over time regardless of how the temperature or humidity in the environment might be changing, the equivalent flux term is zero and thus the equivalent functions are ODEs, which are much easier to solve. Thus, we eliminate $d_e$ as a choice for the diameter parameter.

The model in this manuscript can be constructed with either choice of $d_0$ or $d_D$, with $w$ appearing in places if $d_0$ is chosen, and both $\delta$ and $\zeta$ appearing in places if $d_D$ is chosen. We choose $d_0$ because then we only need one of the ratios ($w$ only), the diameter limits are easier to express in it, and the literature on the diameter distributions of exhaled aerosols generally work hard to convert their measurements (vary between whether they are $d_e$ or $d_D$) into expressions in terms of $d_0$ rather than $d_D$.

Now, $n_k(d_0, t)$ is the concentration density of aerosols in terms of $t$ and the original diameter $d_0$. Let $\tilde{n}_k$ be the concentration density in terms of $t$ and $d_e$, and $\breve{n}_k$ be the concentration density in terms of $t$ and $d_D$. To make conversions between them; consider the original diameter interval $d_0$ to $d_0 + dd_0$, and its corresponding intervals $d_e$ to $d_e + dd_e$ and $d_D$ to $d_D + dd_D$. The number of aerosols in each interval must all be equal: $n_k \, dd_0$, $\tilde{n}_k dd_e$, and $\breve{n}_k dd_D$. Thus, the conversions are

$$\tilde{n}_k = \frac{n_k}{w} , \tag{23}$$

$$\breve{n}_k = \frac{n_k}{\zeta} , \tag{24}$$

$$\tilde{n}_k = \frac{\breve{n}_k}{\delta} . \tag{25}$$

Let $n_{0,k}(d_0)$ be the initial concentration density in the room for a multiplicity $k$ at the initial time $t = t_0$ and $n_{r,k}(d_0, t)$ be the volume averaged concentration density of the air coming in from other rooms. We are assuming that the concentration density outdoors is negligible.

## Diameter limits

For the model, we will limit ourselves for each multiplicity to the range $d_0 \in [d_{m,k}, d_M]$ where $d_{m,k}$ is the minimum aerosol diameter required to hold $k$ pathogen copies, and $d_M$ is a diameter cutoff separating larger aerosols that are more ballistic and gravitationally settle to the ground too quickly to become well mixed and smaller aerosols that more closely follow the flow and mix. Let $K_m(d_0)$ be the largest number of pathogen copies that can fit in an aerosol at production. We will consider

$$n_k(d_0, t) = 0 \ \ \forall \ d \notin [d_{m,k}, d_M], \ k > K_m(d_0) \,. \tag{26}$$

All of these limits have problems, but there is no obvious better choice without adding a lot more complexity to the model.

For a spherical pathogen with diameter $d_p$, we can use the crude approximation of just considering the total pathogen volume and a packing efficiency $e = 0.74$ (hard pack spheres) with a minimum of 1 and completely neglect the aerosol shape that small number of pathogen copies would force (two pathogen copies, for example, can't be arranged into a configuration that even vaguely resembles a sphere). We can use the same idea to get $K_m(d_0)$. Both of them are

$$d_{m,k} \ \approx \ \begin{cases} d_p & \text{if } k = 1 \,, \\ \left(\frac{k}{e}\right)^{1/3} d_p & \text{if } k > 1 \,, \end{cases} \tag{27}$$

$$K_m(d_0) \ \approx \ \max\left[1, e\left(\frac{d_0}{d_p}\right)^3\right] \,. \tag{28}$$

At the lower limit near $d_{m,k}$, the pathogen/s take up a disproportionate amount of the space in the aerosol compared to other solutes and the assumption of approximately equal solute concentrations at production is violated and the evaporation ratio has a strong dependence on $d_0$ and the initial multiplicity, the latter of which we aren't tracking at all. However, as long as the total liquid volume of exhaled aerosols with diameters close to $d_{m,k}$ (say, those whose diameters are small enough that their volume is only a few times larger) is small compared to total liquid volume of the rest of the range in $d_0$, this problem will have a negligible effect. Additionally, the diameter dependence of many of the sink terms may be much smaller close to $d_{m,k}$ for submicron pathogen copies which means that the effect of assuming the wrong evaporation ratio may be small. The smaller the pathogen, the less issues this will pose. It will be least important for small viruses, and possibly quite important for large bacteria and eukaryotic pathogens.

The upper limit is rather imprecise since there is no single hard separation scale that could be chosen unless the air is completely still in which case one can use a so called "Wells curve" (same Wells as of the Wells-Riley model) for the environment's humidity to determine the largest size that won't settle to the ground before evaporating to their equilibrium diameter, such as the original one [24] or newer ones [25]. But mixing of any sort complicates this. One might think that one could just rely on the fact that the gravitational settling sink term keeps growing with diameter and not bother with the problem. But, the well-mixed assumption breaks down and the lifetime of the aerosols converges towards depending solely on the initial

diameter and the height of the infectious individual's mouth and nose from the ground. Additionally, the time to evaporate to the equilibrium diameter increases with increasing size. And from a practical standpoint, it is necessary in order to keep $M_c$ from getting too large since $M_c \sim \mathcal{O}(d_M^3)$ for sufficiently large $d_M$ and pathogen concentration in the infectious individual's respiratory tract fluid $\rho_p$. If we assume that the aerosols are approximately spherical (reasonably true except potentially when completely dried out) and their density is approximately equal to that of water $\rho_w$, the aerosols' inertial response times $\tau_p$ to fluid motions from Stokes drag (we are assuming they are small enough that contributions beyond Stokes drag are negligible) and gravitational settling terminal velocity $u_g$ are

$$\tau_p = \frac{\rho_w d^2}{18 \rho_a v_a} , \tag{29}$$

$$u_g = \frac{(\rho_w - \rho_a) g d^2}{18 \rho_a v_a} \approx g \tau_p , \tag{30}$$

where $\rho_a$ is the density of air, $v_a$ is the kinematic viscosity of air, and $g$ is the acceleration due to gravity.

Both grow quadratically with diameter, which does not lend itself to a well defined cutoff scale. And additionally one must consider that once exhaled, the aerosols will tend to evaporate (relative humidity in the environment is typically lower than in the respiratory tract where it is close to 100%) thereby reducing their inertia and terminal velocities. For 10 μm, 20 μm, and 50 μm diameter aerosols; the terminal velocities at 20°C and atmospheric pressure are 3.0 mm s$^{-1}$, 1.2 cm s$^{-1}$, and 7.5 cm s$^{-1}$ respectively. However, larger aerosols take longer to evaporate/grow to their equilibrium diameter and therefore will settle at a faster rate initially than their final equilibrium diameter suggests, which makes them even more likely to be lost due to settling than smaller aerosols.

The simulations of Chong *et al.* [21] indicate that 100 μm aerosols are quite ballistic and quickly fall out of the exhaled plume, but 10 μm aerosols are carried along with the plume and stay in the air despite their evaporation being greatly slowed. This suggests that $d_M$ should be chosen somewhere in the 10–100 μm range, which is further supported by the Wells curves found by Xie *et al.* [25]. For lack of a better suggestion; we suggest the use of $d_M = 50$ μm, which will be explored in the Discussion. Before evaporating, the terminal velocity is 7.5 cm s$^{-1}$. If the evaporation ratio is a typical value in the $\frac{1}{2} - \frac{1}{5}$ range, the final evaporated diameter would be in the 10–25 μm range and have terminal velocities in the 3–19 mm s$^{-1}$ range which is still in the range that indoor environment air flow can keep suspended (though with a high loss rate).

## People and infectious aerosol production

We will denote infectious individuals by the subscript *I*, susceptible individuals by the subscript *S*, and other individuals by the subscript *O*. The Other category is all the individuals who are non-infectious non-susceptible. This includes individuals that are immune before they enter the environment (following Jimenez [10]), all of the Removed group in SIR and SEIR models except for the individuals who died or leave the environment, and all of the Exposed group in SEIR models. If one wants to make a full SEIR model from the model presented in this manuscript, the two subgroups (Exposed, and the part of Removed that is still within the environment and breathing plus the previously immune individuals) within this group will have to be treated explicitly. Let the number of individuals in category *C* be $N_C$. The total number of individuals is $N = N_I + N_S + N_O$. The subscript *A* will be used to refer to all individuals

in all categories. Each count is potentially a function of time as individuals can come in and out of the environment. Let $\langle \cdot \rangle_C$ denote taking the average over all individuals in category $C$.

Let $\lambda_{C,j}(t)$ be the volumetric breathing rate of the $j$'th person in category $C$. Let $E_{C,m,in,j}(d)$ and $E_{C,m,out,j}(d)$ be the filtering efficiency of the mask (if any) of the $j$'th person in category $C$ for inhalation and exhalation respectively.

The filter efficiencies of most masks vary significantly with aerosol diameter. Note that it is important that the leak rate of the mask be included in its filtering efficiency. These two filtering efficiencies are generally not equal because masks tend to leak more during exhalation than inhalation and aerosols have higher velocities on exhalation than inhalation. We will assume that all infectious aerosols caught by the mask aren't later re-aerosolized.

Let $E_{C,r,j}(d_0, w, \lambda_{C,j})$ be the filtering/absorption efficiency of the respiratory tract of the $j$'th person in category $C$. This term is non-zero, but it is also not equal to one since the respiratory tract does not absorb all infectious aerosols that pass through it [5, 7, 9, 12]. The best example of this is the observation that individuals can inhale smoke (which is composed of many aerosols) and then exhale some of it back out. The filtering efficiency depends on the original diameter of the aerosols and the evaporation ratio in the environment since $d_0$ and $w$ give both the initial diameter on inhalation ($d \approx d_e = w d_0$), the diameter the aerosols grow towards ($d_0$) if they are wet on inhalation or nucleate inside the respiratory tract if they are completely dry on inhalation, as well as the time they spend inside the respiratory tract which is inversely proportional to $\lambda_{C,j}$. It must capture the time it takes for the aerosols to nucleate and grow if they are dried out, the growth process inside the respiratory tract, and the absorption probability as they pass through the respiratory tract. A useful reference for the nucleation and the growth processes would be Pruppacher & Klett [27], and a useful reference for the absorption processes for particles in the respiratory tract would be ICRP [39].

The diameter will be $d_e = wd$ when passing through the mask on inhalation, and $d_0$ when passing through the mask on exhalation since the humidity between the mouth and nose and the mask is high and the distance is short, so there is little time for evaporation. It is often easier to work with the survival efficiencies rather than the filtering efficiencies, defined as

$$S_{C,m,in,j}(d_0, t) \quad = \quad 1 - E_{C,m,in,j}(w(d_0, t)d_0) \,, \tag{31}$$

$$S_{C,r,j,k}(d_0, w, \lambda_{C,j}) \quad = \quad 1 - E_{C,r,j}(d_0, w, \lambda_{C,j}) \,, \tag{32}$$

$$S_{C,m,out,j,k}(d_0) \quad = \quad 1 - E_{C,m,out,j}(d_0) \,. \tag{33}$$

We will assume that the number of infectious pathogen copies in each exhaled droplet/aerosol follow a Poisson distribution where the mean count is equal to the droplet/aerosol's initial volume times the pathogen load in respiratory tract fluid at the point of production. This excludes diseases where pathogenic agents stick together and clump. Note that this implicitly means we are assuming that the pathogen volume fraction in the respiratory tract fluid is small. Otherwise, the non-Poissonity caused by there being a maximum number of pathogen copies that can fit in a finite sized drop will **NOT** be negligible.

Let $\rho_j(d_0, t)dd_0$ be the number density in exhaled air of the aerosols with diameters between $d_0$ and $d_0 + dd_0$ exhaled by the $j$'th infectious individual at time $t$. Let $\rho_{p,j}(t)$ be the pathogen concentration in the $j$'th person's respiratory tract fluid where the aerosols are being produced. The mean/expected multiplicity for infectious aerosols produced by the $j$'th infectious

individual for any $d_0$ is

$$\langle k \rangle (d_0, t)_j = \frac{\pi}{6} d_0^3 \rho_{p,j}(t) \,. \tag{34}$$

If the pathogen copies are Poisson distributed in the fluid that makes up the aerosols (no clumping, etc.), then

$$n_{I,j,k}(d_0, t) = \begin{cases} \rho_j(d_0, t) P_P(\langle k \rangle (d_0, t)_j, k) & \text{if } d_0 \geq d_{m,k} \,, \\ 0 & \text{if } d_0 < d_{m,k} \,. \end{cases} \tag{35}$$

Note that no infectious aerosols with multiplicity $k$ can be generated with diameters too small to contain them (i.e. no $d_0 < d_{m,k}$ aerosols).

## Sources

We will denote sources by the symbol $\beta$ with a subscript denoting the individual source. All of them are normalized by the volume of the environment, $V$.

First, ventilation with other rooms brings infectious aerosols inside at a rate, normalized by the environment volume, of

$$\beta_{r,k}(d_0, t) = q_r(t) n_{r,k}(d_0, t) \,. \tag{36}$$

where we have lumped all other rooms that might be exchanging air with the room of interest together rather than summing over them as done by Noakes & Sleigh [33]. A coupled model for multiple rooms would have to split this into a sum and model the whole system. Note that we are assuming, like elsewhere, the aerosols brought in from other rooms reach their equilibrium diameter quickly compared to other processes.

The other source is the infectious individuals exhaling aerosols with pathogen copies in them. The total production from the infectious individuals normalized by the environment volume is the sum of the products of the breathing rate, the exhaled aerosol concentration density, and the survival efficiency of the mask [7, 10]; which is

$$\begin{aligned}
\beta_{I,k}(d_0, t) &= \frac{1}{V} \sum_{j=1}^{N_I} \overbrace{\lambda_{I,j}(t) n_{I,j,k}(d_0, t)}^{\text{production rate}} \overbrace{[1 - E_{I,m,out,j}(d_0)]}^{\text{mask survival}} \\
&= \frac{N_I}{V} \langle \lambda_I(t) n_{I,k}(d_0, t)[1 - E_{I,m,out}(d_0)] \rangle_I \,,
\end{aligned} \tag{37}$$

where the $j$ subscript has been dropped in the average. Any terms in the average of a product ($\lambda_{I,j}$, $n_{I,j,k}$, and $1 - E_{I,m,out,j,k}$) that have no correlation with the others can be pulled out to make a product of averages. But any correlated terms cannot be separated, which means it must be kept as an average of a product. As an example, if there are two infectious individuals in a room and one is singing and the other is listening in silence; they will be strongly correlated. The singing person will on average be breathing at a higher rate, could have a higher concentration density of infectious aerosols in their exhaled air, and probably won't be wearing a mask while the listener might be wearing a mask. Now, if all individuals are wearing the same mask, the mask term could be pulled out but the other two terms would remain since they could still be correlated.

Other than not replacing the average of the product with the product of the averages, following aerosols by multiplicity and diamater, and not using quanta; this term is identical to

the equivalent term by Nazaroff, Nicas & Miller [7] and Jimenez [10] and, if masks are removed, that of the original formulation [4].

Now, it may be the case that an infectious person has different respiratory tract pathogen concentrations at different locations where exhaled aerosols are produced (e.g. different concentrations in the lungs and mouth). In this case, one would split the term in Eq (37) for the particular infectious person into separate terms for each location of production and use different $\rho_j(d_0, t)$ and $\langle k \rangle (d_0, t)_j$ in $n_{I,j,k}(d_0, t)$ from Eq (35).

## Sinks

Sinks are proportional to the concentration density $n_k$. We will denote all sinks divided the concentration density by the symbol $\alpha$ with a subscript denoting the individual source. All of them are normalized by the volume of the environment, $V$. Unlike the sources, none of the sinks (except inactivation, considered separately) depend on the multiplicity and therefore the subscript $k$ is dropped. Note that inactivation is treated separately later since it is a flux term when considering each multiplicity separately, unlike in the traditional formulation where it is a sink.

The volume normalized loss rate coefficients of infectious aerosols due to exchange of clean air with outdoors and other rooms are just the volume normalized flow rates [9, 33] and are

$$\alpha_o(t) \quad = \quad q_o(t) \,, \tag{38}$$

$$\alpha_r(t) \quad = \quad q_r(t) \,, \tag{39}$$

respectively.

Let $E_v(d)$ be the filtering efficiency of the circulating ventilation system for aerosols with diameter $d$. The diameter when an aerosol reaches this filter is $d \approx d_e = w(d_0, t)d_0$. Then the volume normalized loss rate coefficient from the circulating ventilation system [4] is

$$\alpha_v(d_0, t) = q_v(t)E_v(w(d_0, t)d_0) \,. \tag{40}$$

Aerosols also gravitationally settle and deposit onto surfaces. We will treat these processes as simple loss rates proportional to their concentration densities just as one does with radioactive decay. The volume normalized loss rates divided by the concentration density, of gravitational settling and deposition are defined to be $\alpha_g(w(d_0, t)d_0)$ and $\alpha_d(w(d_0, t)d_0)$ respectively; which depend on the room geometry, aerosol diameter, and air flow in the room. A possible approximate expression for the settling loss term [9] would be

$$\alpha_g(w(d_0, t)d_0) \approx \frac{1}{h}u_g(w(d_0, t)d_0) \,, \tag{41}$$

where $h$ is the characteristic height of the indoor environment and $u_g(d)$ is the terminal velocity. For small spherical aerosols, Eq (30) provides $u_g(d)$. Larger aerosols need additional diameter corrections [9, 25, 40].

## Sinks from individuals inhaling aerosols

Unfortunately, when individuals inhale infectious aerosols, some are absorbed thereby causing a risk of infection. While this phenomena is not desired for susceptible individuals, we must consider the loss rate from this process by the susceptible individuals as well as the infectious individuals and the non-infectious non-susceptible individuals. There are three steps to the filtering process for the $j$'th person of category $C$: passing through the mask on inhalation, passing through the respiratory tract, and then passing through the mask on exhalation.

The total survival probability of an aerosol going through all three steps is the product of the individual survival rates. The total filtering efficiency is then one minus the total survival rate. But, there is a time delay between when the aerosols are removed from the environment on inhalation and when the survivors are exhaled back out. As long as this time is short compared to all other time scales such as mixing times in the room, the time scales of all other sinks, the time scale of inactivation, etc.; we can ignore this time delay and consider the re-exhalation to occur at the same time. This assumption implies that we can neglect possible changes in multiplicity by inactivation while the aerosols are in the respiratory tract. In most situations, this is a reasonably good assumption. But, at a swimming pool where people regularly hold their breath for long periods of time, this assumption could be violated for the highest multiplicities since the inactivation rate from $k$ to $k-1$ is proportional to $k$.

We assume that the individuals are far enough away from sources and that the environment is well-mixed enough that the concentration density in the air inhaled by each individual is approximately the average concentration density $n_k(d_0, t)$. See the Discussion for a brief qualitative discussion of what the required corrections would look like when this assumption is not valid. Note that we will make the assumption that the self-proximity correction for infectious individuals is negligible (each infectious individual is by definition in close proximity to an infectious individual, themself), though this could pose an issue when the transport of infectious aerosols in the environment to an individual is weak [29]. Then the number of aerosols that are inhaled by a person is equal to $\lambda_{C,j}(t) n_k(d_0, t)$. The volume normalized sink coefficient from this filtering is then

$$
\begin{aligned}
\alpha_{C,f}(d_0, t) &= \frac{1}{V} \sum_{j=1}^{N_C} \overbrace{\lambda_{C,j}(t)}^{\text{volume rate}} \Big[ 1 - \overbrace{\underbrace{S_{C,m,in,j}(d_0, t)}_{\text{mask in}} \underbrace{S_{C,r,j,k}(d_0, w, \lambda_{C,j})}_{\text{resp. tract}} \underbrace{S_{C,m,out,j,k}(d_0)}_{\text{mask out}}}^{\text{total filtering efficiency}} \Big] \\
&= \frac{N_C}{V} \langle \lambda_C(t) \Big\{ 1 - \Big[ 1 - E_{C,m,in,j}(w(d_0, t) d_0) \Big] \\
&\quad \bullet [1 - E_{C,r}(d_0, w(d_0, t), \lambda_C(t))][1 - E_{C,m,out}(d_0)] \Big\} \rangle_C ,
\end{aligned}
\tag{42}
$$

where the $j$ subscript has been dropped in the average over category $C$. As was the case before with the average of a product, only terms that are uncorrelated with the others can be pulled out or be replaced by their average value inside. Note that if aerosols completely dry out in the environment, we have made the assumption that their diameters have approximately returned to $d_0$ upon leaving the respiratory tract at re-exhalation. This assumption only effects the value of $\alpha_{C,f}(d_0, t)$ if an individual is wearing a mask.

## Flux: Inactivation

When a pathogen in an aerosol with multiplicity $k$ inactivates, the aerosol's multiplicity changes to $k-1$. We will model inactivation of pathogen copies as exponential decay with inactivation rate $\gamma(t)$, which might depend on time (e.g. dependence on UV light intensity, humidity, etc. that could be fluctuating in time). For aerosols with a multiplicity of $k$, the volume normalized loss rate to multiplicity $k-1$ is just

$$
f_{k,k-1}(t) n_k(d_0, t) = k\gamma(t) n_k(d_0, t) .
\tag{43}
$$

Two pathogen copies will never inactivate at exactly the same time; so we don't have to consider flux terms beyond the two neighboring multiplicities.

## General concentration density equations

All of the sources, sinks, and flux terms can be collected to make the system of differential equations describing the infectious aerosol concentration density, which is

$$\frac{dn_k}{dt} = -\alpha(d_0, t)n_k + f_{k+1,k}(d_0, t)n_{k+1} - f_{k,k-1}(d_0, t)n_k + \beta_k(d_0, t) \,. \tag{44}$$

We have assumed that shattering and collisional coalescence of infectious aerosols, whether from turbulent induced collisions or differential gravitational settling, is negligible. Collisional coalescence could begin to be important if there are a significant number of very large aerosols and/or $n_k$ is very large. Particularly, $d > 100$ μm aerosols/droplets, even though they will generally settle to the ground/floor before evaporating to their equilibrium diameter [24, 25], can capture smaller aerosols on their way to the ground/floor [26, 27, 38]. This will generally be negligible unless individuals are situated in the environment such that the large aerosols exhaled by one person (who need not be infectious) will fall through the exhaled aerosol plume of an infectious individual, and potentially negligible even then. If the aerosol concentration, including non-infectious aerosols, reach the levels seen in atmospheric clouds, collisional coalescence might also have to be considered along with keeping track of $k = 0$ aerosols; though this is very unlikely in indoor environments except when there is a lot of smoke or artificial fog machines are in use, like in a discotheque or theater.

Then, putting the flux terms into Eq (44), we have the following system of ODEs to get the concentration density

$$\frac{dn_k}{dt} = -\alpha(d_0, t)n_k + (k+1)\gamma(t)n_{k+1} - k\gamma(t)n_k + \beta_k(d_0, t) \,. \tag{45}$$

Luckily this is a system of ODEs rather than PDEs with flux terms in diameter (involving derivatives with respect to diameters). This is the advantage of choosing $d_0$ or $d_D$ instead of $d_e$. For practical applications, this also means that we can also split the diameter range into bins and solve it for each bin separately since there are no flux terms between bins. (See S3 Appendix for how to bin the model with respect to diameter.).

This is a linear inhomogeneous finite system of coupled ODEs at each $d_0$. The number of equations in the system is finite since $k$ is non-negative and there is the maximum theoretical multiplicity $M$. Moreover, we don't even need to care about $k = 0$ since those aerosols are no longer an infection hazard. Additionally, the system that needs to be solved is smaller if $M_c < M$. If $M_c = 1$, then we have only one ODE. This situation occurs if the pathogen load of respiratory tract fluid is low enough that very few aerosols have 2 or more pathogen copies in them.

Note that this model demonstrates superposition with respect to sources since it is linear, as expected intuitively—each aerosol is independent of all others, therefore the response (concentration density and expected dose) from each individual source is independent of all other sources. If $n_{k,1}$ and $n_{k,2}$ are solutions for the same $\alpha$ and $\gamma$ but different sources $\beta_{k,1}$ and $\beta_{k,2}$ respectively, then the solution for $\beta_k = \beta_{k,1} + \beta_{k,2}$ is $n_k = n_{k,1} + n_{k,2}$.

## Infection risk

Let $\mu_{j,k}$ be the average number of aerosols with multiplicity $k$ absorbed by the $j$'th susceptible individual from time $t_0$ to time $t$. At any particular instant of time, the average number of such aerosols of each original diameter $d_0$ entering the person's mask if they are wearing a mask or their mouth and nose if they aren't is $\lambda_{S,j}(t)n_k(d_0, t)$. Note that we have assumed that the $j$'th susceptible individual is not close enough to any sources or filtering sinks that the concentration density of the air they are inhaling deviates significantly from $n_k(d_0, t)$. For susceptible

individuals in close proximity to infectious individuals, close to the output of ventilation, etc.; corrections must be applied. See the Discussion for a qualitative discussion on what the required corrections would look like.

A fraction $S_{S,m,in,j}(d_0, t)$ will survive the mask to enter the respiratory tract [5–8, 10, 12]. A fraction $E_{S,r,j}(d_0)$ of those survivors will be absorbed by the respiratory tract [5, 7, 9, 12], which contributes to the dose. The expected average **aerosol** dose is then the double integral of this over the $d_0$ and the time between $t_0$ and $t$, which is

$$
\begin{aligned}
\mu_{j,k}(t) &= \int_{d_{m,k}}^{d_M} d\phi \int_{t_0}^{t} dv \overbrace{E_{S,r,j}(\phi, w(\phi, v), \lambda_{S,j}(v))}^{\text{absorption efficiency}} \overbrace{S_{S,m,in,j}(\phi, t)}^{\text{survive mask}} \overbrace{\lambda_{S,j}(v) n_k(\phi, v)}^{\text{inf. aerosol inhalation rate}} \\
&= \int_{d_{m,k}}^{d_M} d\phi \int_{t_0}^{t} dv \left\{ E_{S,r,j}(\phi, w(\phi, v), \lambda_{S,j}(v)) \right. \\
&\quad \left. \bullet [1 - E_{S,m,in,j}(w(\phi, v)\phi)]\lambda_{S,j}(v) n_k(\phi, v) \right\},
\end{aligned}
\tag{46}
$$

where we have $\phi$ as the integration variable over $d_0$ and $v$ as the integration variable over time. We will continue to use $\phi$ and $v$ exclusively for this purpose in the rest of the manuscript.

In order to use the $\mu_{j,k}$ in the multiplicity-corrected dose-response model for the particular disease of interest $\mathfrak{R}$, we need to first assume that the **aerosol** dose for each multiplicity follows a Poisson distribution with $\mu_{j,k}$ as the means and that each is independent of each other (no correlations). This requires the well-mixed assumption like many other parts of the model.

But it also requires that the effect of turbulent inertial clustering is negligible. We will now show that it is negligible except possibly at extremely high aerosol concentrations. It will be negligible if the aerosol Stokes numbers $St = \tau_p/\tau_\eta$ are very small ($St \ll 1$) [41, 42] where $\tau_p$ is the aerosol inertial response time scale from Eq (29) and $\tau_\eta$ is the Kolmogorov time scale of the turbulence in the environment, which is $\tau_\eta = \sqrt{v_a/\epsilon}$ where $\epsilon$ is the turbulent dissipation rate. It will also be small if the typical inter-aerosol distance $\bar{d}_a \sim \mathcal{N}^{-1/3}$, where $\mathcal{N}$ is the total infectious aerosol concentration for all $d_0$ and $k$, is much larger than the typical scale of turbulent inertial clustering (i.e. the fraction of aerosols with a neighbor in the clustering range is low). The typical scale of turbulent inertial clustering is about $10\eta$ [41, 42] where $\eta = (v_a^3/\epsilon)^{1/4}$ is the Kolmogorov length scale of the turbulence. This means that as long as $St \ll 1$ and/or $\mathcal{N}^{-1/3} \gg 10\eta$, the deviations of the aerosol doses from independent Poisson distributions will be negligible. The situation will be worst for the largest $w(d_M, t)d_M$ sized aerosols in high enough humidity that $w(d_M, t) \approx 1$. For a low dissipation rate of $\epsilon = 1$ mW kg$^{-1}$; $St = 0.06$ for a $d_M$ sized aerosol and the number density limit is $\mathcal{N} \ll 4 \times 10^5$ m$^{-3}$. The Stokes number is small, so the turbulent inertial clustering's effect will be small even if $\mathcal{N}$ exceeded that limit. For a higher dissipation rate of $\epsilon = 1$ W kg$^{-1}$; $St = 2.0$ for a $d_M$ sized aerosol and the number density limit is $\mathcal{N} \ll 7 \times 10^7$ m$^{-3}$. While the Stokes number is large, the number density limit is very high so turbulent inertial clustering's effect will generally be small. For a high for indoors dissipation rate of $\epsilon = 10$ W kg$^{-1}$; $St = 6.3$ for a $d_M$ sized aerosol and the number density limit is $\mathcal{N} \ll 4 \times 10^8$ m$^{-3}$. While the Stokes number is large, the number density limit is very high so turbulent inertial clustering's effect will generally be small. Thus, turbulent inertial clustering will have a negligible effect on the Poissonity and independence of the aerosol dose distributions except possibly at extraordinarily high aerosol concentrations.

## Model solution and simplification

### General

There is an analytical solution to Eq (45), though it is not closed form unless the time dependence of $\alpha$, $\beta$, and $\gamma$ allow it. Eq (45) can be rewritten in matrix-vector form as

$$\frac{d\vec{n}}{dt} = \mathbf{A}(d_0, t)\vec{n}(d_0, t) + \vec{\beta}(d_0, t), \tag{47}$$

where $\vec{n}(d_0, t)$ and $\vec{\beta}(t)$ are the $n_k(d_0 t)$ and $\beta_k(d_0, t)$ for $k > 0$ in vector form and

$$\mathbf{A} \equiv \begin{bmatrix} -\alpha(d_0, t) - \gamma(t) & 2\gamma(t) & & & \\ & -\alpha(d_0, t) - 2\gamma(t) & 3\gamma(t) & & \\ & & \ddots & \ddots & \\ & & & \ddots & M_c\gamma(t) \\ & & & & -\alpha(d_0, t) - M_c\gamma(t) \end{bmatrix}. \tag{48}$$

is an upper bidiagonal $M_c \times M_c$ square matrix. For any fixed $d_0$ or bin of $d_0$, the resulting system of ODEs is particularly amenable to efficient numerical solution even for very large $M_c$ because A is sparse with only one or two elements per row.

The general solution in matrix-vector form, shown in S1 Appendix, is

$$\vec{n}(d_0, t) = \exp\left[\int_{t_0}^{t} A(d_0, x)dx\right]\vec{n}_0(d_0) + \int_{t_0}^{t} \exp\left[\int_{s}^{t} A(d_0, x)dx\right]\vec{\beta}(d_0, s)ds. \tag{49}$$

Working this out using the structure of the diagonalization of **A** in S1 Appendix, the general solution for each $k$ is

$$\begin{aligned} n_k(d_0, t) = \exp&\left[-\int_{t_0}^{t} \alpha(d_0, x)dx\right]\exp\left[-k\int_{t_0}^{t} \gamma(x)dx\right] \\ &\bullet\sum_{p=k}^{M_c}\binom{p}{k}n_{0,p}(d_0)\left[1 - \exp\left[-\int_{t_0}^{t} \gamma(x)dx\right]\right]^{p-k} \\ &+\sum_{p=k}^{M_c}\binom{p}{k}\int_{t_0}^{t}\beta_p(d_0, s) \\ \bullet\exp&\left[-\int_{s}^{t}\alpha(d_0, x)dx\right]\exp\left[-k\int_{s}^{t}\gamma(x)dx\right]\left[1 - \exp\left[-\int_{s}^{t}\gamma(x)dx\right]\right]^{p-k}ds, \end{aligned} \tag{50}$$

where $\binom{k}{m} = k!/(m!(k-m)!)$ is the notation for the binomial coefficient $k$ choose $m$.

### Coefficients constant in time

We cannot go further in simplifying the general solution from Eq (50) without knowing the time dependence of $\alpha$, $\vec{\beta}$, and $\gamma$. In many situations; $\alpha$, $\vec{\beta}$, and $\gamma$ are approximately constant with respect to time. If this is so; the general solution from Eq (50) and its time integral from $t_0$

to $t$ (needed for the dose) for the trivial case that $\gamma = 0$ but $\alpha \neq 0$ is

$$\vec{n}_\infty \quad = \quad \frac{1}{\alpha}\vec{\beta} \,, \tag{51}$$

$$\vec{n} \quad = \quad \vec{n}_\infty + (\vec{n}_0 - \vec{n}_\infty)e^{(t-t_0)\alpha} \,, \tag{52}$$

$$\int_{t_0}^{t} \vec{n}(v)dv \quad = \quad (t - t_0)\vec{n}_\infty + \frac{1}{\alpha}(\vec{n}_0 - \vec{n}_\infty)\left[1 - e^{(t-t_0)\alpha}\right] . \tag{53}$$

For the trivial case that both $\gamma = 0$ and $\alpha = 0$, the solution is instead

$$n_{\infty,k} \quad = \quad \begin{cases} 0 & \text{if } \beta_k = 0 \,, \\ +\infty & \text{otherwise} \,, \end{cases} \tag{54}$$

$$\vec{n} \quad = \quad \vec{n}_0 + (t - t_0)\vec{\beta} \,, \tag{55}$$

$$\int_{t_0}^{t} \vec{n}(v)dv \quad = \quad (t - t_0)\vec{n}_0 + \frac{1}{2}(t - t_0)^2\vec{\beta} \,. \tag{56}$$

But for the general case of $\gamma \neq 0$, the solution is instead (see S1 Appendix)

$$n_k(d_0, t) \quad = \quad n_{\infty,k} + z^s[U_k(d_0, \vec{\beta}(d_0), z) + V_k(\vec{n}_0(d_0), z)] \,, \tag{57}$$

$$\int_{t_0}^{t} n_k(d_0, v)dv \quad = \quad (t - t_0)n_{\infty,k}(d_0)$$
$$- U_k(d_0, \vec{n}_0(d_0), 1) + z^s U_k(d_0, \vec{n}_0(d_0), z) \tag{58}$$
$$- \frac{1}{\gamma}W_k\left(d_0, \vec{\beta}, z\right) \,,$$

where

$$z(t) \quad = \quad e^{-(t-t_0)\gamma} \in (0, 1] \,, \tag{59}$$

$$s(d_0) \quad = \quad \frac{\alpha(d_0)}{\gamma} + k \,, \tag{60}$$

$$V_k(\vec{y}, x) \quad = \quad \sum_{i=k}^{M_c} \binom{i}{k}y_i(1 - x)^{i-k} \,, \tag{61}$$

$$U_k(d_0, \vec{y}, x) \quad = \quad -\frac{1}{\gamma}\sum_{i=k}^{M_c}\binom{i}{k}y_i\sum_{p=0}^{i-k}\binom{i-k}{p}\frac{(-1)^p x^p}{s+p} \,, \tag{62}$$

$$W_k(d_0, \vec{y}, x) \quad = \quad \int_{1}^{x} dv\, v^{s-1} U_k(d_0, \vec{y}, v) \,, \tag{63}$$

$$= -\frac{1}{\gamma}\sum_{i=k}^{M_c}\binom{i}{k}\beta_i(d_0)\sum_{p=0}^{i-k}\binom{i-k}{p}\frac{(-1)^p(z^{s+p}-1)}{(s+p)^2}\,,\tag{64}$$

and $n_{\infty,k}(d_0)$ is the concentration density as $t\to\infty$ which is

$$n_{\infty,k}(d_0)=-U_k\left(d_0,\vec{\beta},1\right)=\frac{1}{\gamma}\sum_{i=k}^{M_c}\binom{i}{k}\beta_i(d_0)\sum_{p=0}^{i-k}\binom{i-k}{p}\frac{(-1)^p}{s+p}\,,\tag{65}$$

Note that $s$ is a function of $k$ and $e^{-(\alpha+k\gamma)(t-t_0)}=z^s$.

It is possible for $\lambda_{S,j}$ to be a function of $t$ but $\alpha$ not be (i.e. there is cancelation). But if $\lambda_{S,j}$ and $w$ are constant, the expected average **aerosol** dose of multiplicity $k$ for the $j$'th susceptible individual in Eq (46) becomes

$$\mu_{j,k}(t)=\lambda_{S,j}\int_{d_{m,k}}^{d_M}d\phi E_{S,r,j}(\phi,w,\lambda_{S,j})(1-E_{S,m,in,j}(w\phi))\int_{t_0}^t n_k(\phi,v)dv\,.\tag{66}$$

Calculation of $\vec{n}_k(d_0,t)$, $\vec{n}_{\infty,k}(d_0)$, $\int_{t_0}^t n_k(d_0,v)dv$ scales as $\mathcal{O}(M_c^3)$ due to there being $M_c$ multiplicities and double sums in $U_k$ and $W_k$ that scale as $M_c$. There is a recursive solution for $\vec{n}_{\infty,k}(d_0)$ which is linear in $M_c$, and recursive solutions for all the $U_k$ and $W_k$ which are quadratic in $M_c$. Additionally, the recursive formulas don't require as much numerical precision in the intermediate steps to get a desired final precision as shown in S5 Appendix. From S1 Appendix, the recursive solutions start at $k=M_c$ and proceed downwards to $k=1$. They are

$$U_k(d_0,\vec{y},x)=\begin{cases}-\dfrac{y_{M_c}}{\gamma s}&\text{if }k=M_c\,,\\[2mm]\dfrac{(k+1)x}{s}U_{k+1}(d_0,\vec{y},x)-\dfrac{1}{\gamma s}V_k(\vec{y},x)&\text{otherwise}\,,\end{cases}\tag{67}$$

$$W_k(d_0,\vec{y},x)=\begin{cases}\dfrac{y_{M_c}}{\gamma s^2}\left(1-x^s\right)&\text{if }k=M_c\,,\\[2mm]\dfrac{1}{s}[(k+1)W_{k+1}(d_0,\vec{y},x)\\[1mm]\quad+x^sU_k(d_0,\vec{y},x)-U_k(d_0,\vec{y},1)]&\text{otherwise}\,,\end{cases}\tag{68}$$

$$U_k(d_0,\vec{y},1)=\begin{cases}-\dfrac{y_{M_c}}{\gamma s}&\text{if }k=M_c\,,\\[2mm]\dfrac{(k+1)x}{s}U_{k+1}(d_0,\vec{y},1)-\dfrac{y_k}{\gamma s}&\text{otherwise}\,,\end{cases}\tag{69}$$

$$n_{\infty,k}=\begin{cases}\dfrac{\beta_{M_c}}{\gamma s}&\text{if }k=M_c\,,\\[2mm]\dfrac{1}{\gamma s}\left[\beta_k+(k+1)\gamma\,n_{\infty,k+1}\right]&\text{otherwise}\,.\end{cases}\tag{70}$$

This recursive analytical solution for $\vec{n}$ is checked against a numerical solution of Eq (47) for a simple case and a very small time step in S2 Appendix. The relative differences for the simple case are very small at less than $10^{-12}$. See S5 Appendix for numerical considerations for evaluating the analytical solutions on a computer or solving Eq (47) with a numerical ODE solver. The number of terms for both are discussed, as well as the required precision and maximum magnitude required for floating point numbers used to calculate the analytical solution formulas.

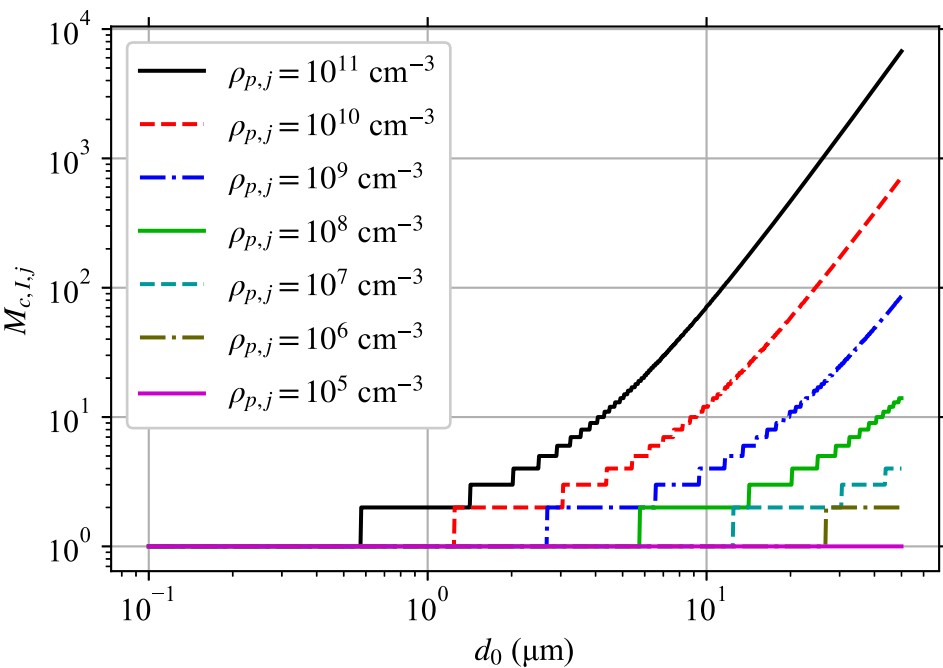

**Fig 2. Required $M_c$ based on pathogen concentration in infectious individuals.** $M_{c,I,j}$ required to capture 99% of pathogen production for each diameter at aerosol production $d_0$ from an infectious individual, with each line being a different pathogen concentration in their respiratory tract fluid $\rho_{p,j}$ (see legend).

## Determining the cutoff Mc

In order to reduce the number of equations that have to be solved, we need to find a suitable cutoff $M_c < M$ if at all possible, whether for the whole diameter range or for each diameter bin (advantage of doing a separate one for each bin is that $M_c$ tends to be small for the small diameter bins), such that the contribution of all higher multiplicities is less than a threshold $T \in (0, 1]$ fraction of the total contribution from all multiplicities. In many cases, this depends only on the $\rho_{p,j}$ of the infectious individuals and one can skip directly to Eq (80) for the value of $M_c$ to use (shown in Fig 2 for a few $\rho_{p,j}$). However, some cases such as when one starts the model after some number of infectious individuals have left the environment, when there is significant transport from other rooms, etc. require additional heuristics. These heuristics are developed below.

A cutoff is suitable if the total contribution for all $k > M_c$ to the average pathogen dose and therefore infection risk is small compared to the total contribution for $k \leq M_c$. It is almost always true that $M_c < M$, and in many cases it can even be $M_c = 1$. This depends on the distribution of exhaled aerosol sizes and the pathogen concentration $\rho_p$ in the respiratory tract fluid where the aerosols are produced. For very low pathogen loading, one can use $M_c = 1$. Let $d_-$ and $d_+$ be the bounds in $d_0$ of the bin (or whole range in which case $d_- = d_{m,1}$ and $d_+ = d_M$) being considered.

The most reliable way to determine $M_c$ is to use the model with the cutoff $M$ and determine $M_c$ afterwards using the result, but that defeats the point of finding $M_c$ since the effort one wants to save has already been expended. So we need heuristics to determine $M_c$ ahead of time. All of them consider the dose contribution from high multiplicity aerosols and consider a simplified $k\mu_{j,k}$ from Eq (46) with a particular concentration density multiplied by the average absorption efficiency of susceptible individuals. For each heuristic, we will define this

parameter to be $\mathcal{H}_{h,k}(t)$ where the $h$ denotes the particular heuristic. Then, the heuristic for $M_c$ is that we must find the $M_c$ such that

$$\sum_{k=1}^{M_c} \mathcal{H}_{h,k}(t) \gg \sum_{k=M_c+1}^{\infty} \mathcal{H}_{h,k}(t) \;\; \forall \;\; h, t \geq t_0 \,. \tag{71}$$

Note that we must take the largest $M_c$ out of the values suggested by the individual heuristics.

An equivalent way to express this heuristic is to look at the ratio of the sum of $\mathcal{H}_{h,k}$ after the cutoff ($k > M_c$) to the total, defined as

$$J_{h,M_c}(t) \equiv \frac{\sum_{k=M_c+1}^{\infty} \mathcal{H}_{h,k}(t)}{\sum_{k=1}^{\infty} \mathcal{H}_{h,k}(t)} \,. \tag{72}$$

Now, $J_{h,M_c}(t) \in [0, 1]$ and is approximately the ratio of the contribution of the higher multiplicities $k > M_c$ aerosols to the total, which we want to be small. An equivalent statement of the heuristics is that one must find the $M_c$ such that $J_{h,M_c} \ll 1 \; \forall \; h, t \geq t_0$. One way to determine $M_c$ is to say pick some threshold $T \in (0, 1]$, and then find the smallest $M_c$ such that $J_{h,M_c} \leq T$ for all heuristics. Let $M_{c,h}(T)$ be the smallest value of $M_c$ that satisfies $J_{h,M_{c,h}}(t) \leq T$, which makes it the single heuristic value of $M_c$. Then, $M_c$ is just the maximum $M_{c,h}$.

First, we define the average absorption efficiency of the susceptible individuals as

$$A_S(d_0, t) \equiv \langle E_{S,r}(d_0, w(d_0, t), \lambda_S)[1 - E_{S,m,in}(w(d_0, t)d_0)]\rangle_S \,. \tag{73}$$

If the $\alpha$, $\beta$, $\gamma$, and $w$ are constant in time; it is a lot less effort to calculate $n_{\infty,k}(d_0)$ using Eq (65) than $n_k(d_0)$. Then, each $\mu_{j,k} \sim A_S \, n_{\infty,k}$. If $q_r(t)$ and $n_{r,k}(d_0, t)$ are non-zero, the doses from them have a similar scaling. If the initial concentration density includes a lot of aerosols with high multiplicities, we will need to set $M_c$ to be large enough to include them even if they won't matter after the initial time. We need to consider this if $n_{0,k} \gg n_{\infty,k}$ for any $k > 1$, and they will have a similar scaling. These heuristics are

$$\mathcal{H}_{\infty,k} \;\; = \;\; k \int_{d_-}^{d_+} A_S(\phi, t) n_{\infty,k}(\phi) d\phi \,, \tag{74}$$

$$\mathcal{H}_{r,k}(t) \;\; = \;\; k \int_{d_-}^{d_+} A_S(\phi, t) n_{r,k}(\phi, t) d\phi \,, \tag{75}$$

$$\mathcal{H}_{0,k}(t) \;\; = \;\; k \int_{d_-}^{d_+} A_S(\phi, t) n_{0,k}(\phi, t) d\phi \,. \tag{76}$$

The last heuristic is similar but considers the infectious individuals inside the environment instead of the concentration density. This has the advantage of not needing to determine $n_{\infty,k}(d_0)$. We essentially take the average over the $d_0$ interval of $\beta_{I,k}(d_0)$ from Eq (36) times the absorption efficiency of the average susceptible individual. We thus define the infectious individuals heuristic parameter

$$\mathcal{H}_{I,k}(t) \equiv k \int_{d_-}^{d_+} d\phi \, A_S(\phi, t) \sum_{j=1}^{N_I} \lambda_{i,j}(t) n_{I,j,k}(\phi, t)[1 - E_{I,m,out,j}(\phi)] \,. \tag{77}$$

But there are practical difficulties in using it directly. So instead, we will define the heuristic for each individual infectious individual using the largest diameter in the range $d_+$, and one

would use the maximum $M_c$ indicated by all of these. This has the advantage that there is a simple form for the required $M_c$, which is derived in S4 Appendix. It is

$$M_{c,I,j}(d_+, T) = 1 + C_P^{-1}(\langle K\rangle(d_+, t)_j, (1 - T)C_P(\langle K\rangle(d_+, t)_j, K_m(d_+) - 1)) \,, \tag{78}$$

where $C_P$ is the CDF (Cumulative Distribution Function) of the Poisson distribution and $C_P^{-1}(\mu, c)$ is the inverse CDF to find the smallest $k$ for which $C_P(\mu, k) \geq c$. Note that when $K_m(d_+) \gg 1$ and $K_m(d_+) \gg \langle k\rangle(d_+, t)_j \gg 1$, $C_P(\langle k\rangle(d_+, t)_j, K_m - 1) \simeq 1$ and

$$M_{c,I,j}(d_+, T) \simeq 1 + C_P^{-1}(\langle K\rangle(d_+, t)_j, (1 - T)) \,. \tag{79}$$

When the assumptions don't apply, this will give an overestimation, so it is usable to get the value of $M_c$ to use. It will just give a bigger value than necessary.

Fig 2 shows $M_{c,I,j}$ as a function of $d_0$ for several different $\rho_{p,j}$. Increasing $\rho_{p,j}$ approximately just shifts the curves for $M_{c,I,j}$ to the left on a log-scale. Notice the very strong effect of $\rho_{p,j}$ on $M_c$, with values a little under 7000 being required for the largest diameter bin for $\rho_{p,j} = 10^{11}$ cm$^{-3}$ and a value of 2 being required for the same bin for $\rho_{p,j} = 10^6$ cm$^{-3}$. Since $M_c$ increases with $d_0$, the vast majority of the effort to determine the concentration density and the infection risk will be spent on the largest bins except for small values of $\rho_{p,j}$.

## Example for SARS-CoV-2 with high viral load

### Room, people, and filter efficiencies

We consider a hypothetical example based on the ongoing SARS-CoV-2 pandemic—a poorly ventilated seminar room with two infectious individuals with SARS-CoV-2 at the very upper end of viral concentrations (viral load) and one of them continuously coughing. We assume that the room is well-mixed and that the individuals are far enough apart from each other and the ventilation that no corrections to $n_k(d_0, t)$ need to be applied at any source or sink, nor in the calculated absorbed doses. Let the room have volume $V = 200$ m$^{-3}$ with a height of $h = 4$ m, with ventilation $q_r = 0$, $q_v = 0$, and $q_o = 0.5$ hr$^{-1}$. We will ignore surface tension's effects on $w$. Let the humidity be such that the evaporation ratio is $w = \frac{1}{3}$, which is a constant with respect to both $t$ and $d_0$. We ignore deposition ($\alpha_d = 0$). Let there be $N_S = 15$ susceptible individuals in groups of 5 wearing no mask, a simple1 mask, and a simple2 mask (defined later); and no non-infectious non-susceptible individuals ($N_O = 0$). The susceptible individuals will be assumed to be sedentary/passive adults with a breathing rate of $\lambda_{S,j} = 0.3$ m$^{-3}$ hr$^{-1}$, which is in the range of mean breathing rates for this activity from the U.S. EPA's *Exposure Factors Handbook* Table 6.2 [14]. The pathogen concentration for SARS-CoV-2 varies widely across individuals, location in the body, and stage of the disease [35, 36, 43, 44], and can sometimes get as high as the $10^{10}$–$10^{11}$ cm$^{-3}$ range [35, 36]. We will use this upper range because it makes the model more challenging to solve due to the larger $M_c$ and due to the interest in so called "super-spreading events". The situation is composed of two stages (Stages 1 and 2) that each start when an infectious individual enters the room. Initially, there are no infectious aerosols in the room, meaning $n_{0,k}(d_0) = 0$. Stage 1; at $t = t_0 = 0$, one infectious individual enters the room who is speaking, wearing no mask, breathing at a rate $\lambda_{I,j} = 0.5$ m$^{-3}$ hr$^{-1}$ (just below an 0.54 m$^{-3}$ hr$^{-1}$ average value for reading out loud [15]), and has a high respiratory tract fluid pathogen concentration of $\rho_{p,j} = 10^{10}$ cm$^{-3}$. Stage 2; then at $t = 3$ hr, one more infectious individual enters the room who is continuously coughing while wearing a simple2 mask, breathing at a higher rate of $\lambda_{I,j} = 2.0$ m$^{-3}$ hr$^{-1}$, and has a higher respiratory tract fluid pathogen concentration of $\rho_{p,j} = 10^{11}$ cm$^{-3}$ at the very upper range for SARS-CoV-2. We chose this estimated continuous coughing breathing rate by deducing a breathing rate range from Hegland, Troche

& Davenport [17] for continuous 3 cough cycles (heavily using their Fig 1), getting a breathing rate range of 1.9–2.3 m$^{-3}$ hr$^{-1}$ from which we chose 2.0 m$^{-3}$ hr$^{-1}$.

We use mask filter efficiencies of the functional form

$$E_{C,m,in,j}(d) = E_{C,m,out,j}(d) = E_\infty - (E_\infty - E_0)e^{-d/D_{m,c}} , \tag{80}$$

where $E_\infty$ is the aerosol filtering efficiency as $d \to \infty$, $E_0$ is the aerosol filtering efficiency as $d \to 0$, and $D_{m,c}$ is the scale of the mask efficiency transition. We will use $D_{m,c} = 10$ μm. We consider individuals wearing no masks or one of two types of masks. Their filtering efficiencies are

**none (no mask)** $E_0 = E_\infty = 0$.

**mask simple1** $E_0 = 0.2$ and $E_\infty = 0.8$.

**mask simple2** $E_0 = 0.95$ and $E_\infty = 0.99$.

The filtering efficiencies of both the simple1 and simple2 masks are shown in S1 Fig. The mask parameters were chosen such that they are more efficient at filtering large aerosols/droplets than small ones, with the simple2 mask being better than the simple1 mask. The simple1 and simple2 masks could reasonably correspond to a reasonably well fitted home-made cloth mask and an excellently fitted FFP2 mask, though here we have treated their leak rate to be the same during inhalation as exhalation (not true with most real masks). At the largest sizes, leakage doesn't matter as much since the aerosols are more ballistic. Let us assume that $E_{C,r,j}(d_0, w(d_0,t), \lambda_{C,j}(t)) = \frac{1}{2}$ for everyone.

## Disease and infectious aerosol production

We assume that an exponential-dose response model is the correct model to use for SARS-CoV-2 since the exponential model works better than the beta-Poisson model for two other human infecting corona viruses (SARS-CoV-1 and HCoV-229E) [34]. In absence of a good value to use for $r$, we use the same value of $r$ as found for SARS-CoV-1 in mice which is $r = 2.45 \times 10^{-3}$ and the same value of $r$ as found for HCoV-229E in humans which is $r = 5.39 \times 10^{-2}$ [34]. We use $\gamma = 0.64$ hr$^{-1}$ as the inactivation rate for SARS-CoV-2 [45].

We approximate the SARS-CoV-2 pathogen as a sphere with a diameter of 100 nm, which is close to the correct size and the rough shape with the surface proteins removed (actually an ellipsoid) [37]. We use the aerosol size distributions for speaking and coughing from Johnson *et al.* [22], but extrapolate them to smaller diameters (from 800 nm to 100 nm). This is used with Eqs (26) and (27) to get the $\beta_{I,k}$. They are shown in the top-right panel of Fig 3. The aerosol size distributions have two peaks at approximately 2 μm and 100 μm. This puts $d_M$ between the trough (between the two peaks) and the second larger diameter peak.

## Concentration densities and infection risk

We now find the infectious aerosol concentration densities and doses, and mean infection risks $\mathfrak{R}_E$. First, we split the diameter range between $d_{m,1} = 0.1$ μm and $d_M = 50$ μm into 20 logarithmically spaced bins; and determine the bin average values for the coefficients over each bin by integration following the scheme in S3 Appendix. The infectious individuals source parameters for the $i$'th bin, $\beta_{I,k}|_i$, are calculated numerically via Simpson's rule for integration with 1000 equal linear width sub-bins in each bin. The particular choice of the mask survival efficiency in Eq (81) and $w$ being constant lets the other binning integrals be calculated analytically.

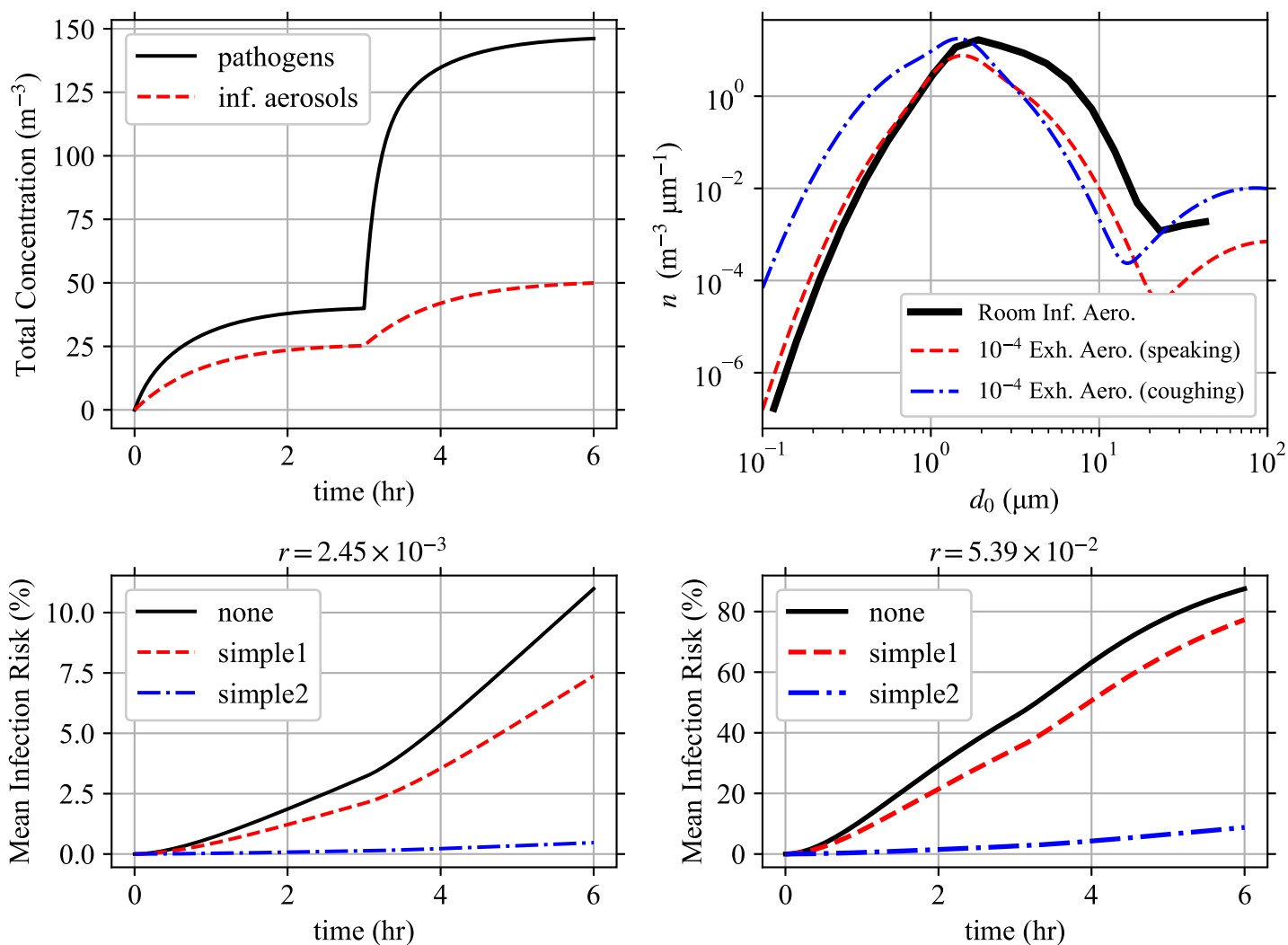

**Fig 3. Model solution for example.** Solution to the example case. (Top-Left) The total pathogen and infectious aerosol concentrations over time. (Top-Right) The infectious aerosol concentration densities in the room as a function of $d_0$ at $t = 6$ hr compared to the aerosol concentration densities being exhaled by speaking and coughing individuals from Johnson *et al.* [22] scaled by $10^{-4}$ to make them have comparable magnitudes. (Bottom-Left, Bottom-Right) The mean infection risk $\mathfrak{R}_E$ for the susceptible individuals based on the mask they are wearing (none, simple1, or simple2) using (Bottom-Left) $r = 2.45 \times 10^{-3}$ (Bottom-Right) $r = 5.39 \times 10^{-2}$.

The model is solved for Stage 1 and then the final values used as initial values for Stage 2 because this makes it so that $\alpha$ and $\beta_k$ are constant in time when solving the model (all changes are between stages). For $M_c$, we used the maximum value of $M_{c,I,j}$ for each infectious individual present at each Stage with $T = 10^{-3}$. Note that $M_c$ stayed the same or increased for each bin going from Stage 1 to Stage 2 with the addition of one more infectious individual.

For the $i$'th bin, the $n_k|_i(t)$ and $\mu_{j,k}|_i(t)$ are solved analytically if $M_c \leq 500$ using the recursive solution and numerically if $M_c > 500$, both in IEEE-754 `binary64` floating point (also known as double precision and `float64`). This threshold between analytical and numerical solving was chosen to use the analytical solution as much as possible without overflow in $V_k$ (see S5 Appendix). As shown in S5 Appendix, `binary64` numbers provide sufficient precision and allowed maximum magnitude. Note that overflow is easy to spot as infinities, which were not seen so this number format was sufficient to prevent overflow. When doing it numerically, Eq (47) along with $\int_0^t n(d_0, v)dv$ were solved using Runge-Kutta 4 with a time step of

$10^{-4}$ hr, which is required for stability and an accurate solution with the large $\alpha|_i + M_c\,\gamma$ values in the largest bin. After determining $\alpha|_i$ and $\vec{\beta}|_i$, the solutions were calculated with the help of the PMADRA (Poly-Multiplicity Airborne Disease Risk Assessment) software suite we wrote for the purpose (https://gitlab.gwdg.de/mpids-lfpn-public/pmadra), specifically the Python 3.5 or newer implementation pypmadra version 0.2.1 (https://gitlab.gwdg.de/mpids-lfpn-public/pmadra/pypmadra) using the Fortran 2008 accelerator library libpmadra version 0.2.1 (https://gitlab.gwdg.de/mpids-lfpn-public/pmadra/libpmadra). The main results are shown in Fig 3.

The total pathogen concentration is slightly less than double the infectious aerosol concentration in Stage 1, and slightly higher than double in Stage 2. This means that the average multiplicity in both stages is approximately two, and it increases slightly from Stage 1 to Stage 2 which is expected with the higher viral load in the second infectious individual. Also, as expected, increasing $r$ (infection risk of each individual pathogen) increases the infection risk. As expected, susceptible individuals wearing masks decrease their infection risk and increasing exposure increases their infection risk.

Comparing the infectious aerosol concentration density in the room with the aerosol concentration densities exhaled by the infectious individuals as a function of $d_0$ (see top-right panel of Fig 3); we can see how as $d_0$ increases, the probability of an aerosol being infectious increases (infectious aerosol concentration density decreases slower after the first peak than the exhaled aerosol concentration densities) but at the largest $d_0 > 15$ μm the increasing $\alpha$ due to stronger gravitational settling causes the infectious aerosol concentration density to grow slower after the trough than the exhaled aerosol concentration densities from the infectious individuals (including the speaking individual who is not wearing a mask). To see the latter, the strengths of the sinks $\alpha$ and total sinks $\alpha + k\gamma$ are shown in Fig 4 and we can see that settling causes $\alpha$ to increase by over a factor of 10 from 100 nm to 50 nm. Fig 4 additionally shows the increase in the total sink strength for the largest multiplicities $M_c$ being considered due to inactivation. The large difference between the total sink strength between $k = M_c$ and $k = 1$ makes the system of ODEs stiff.

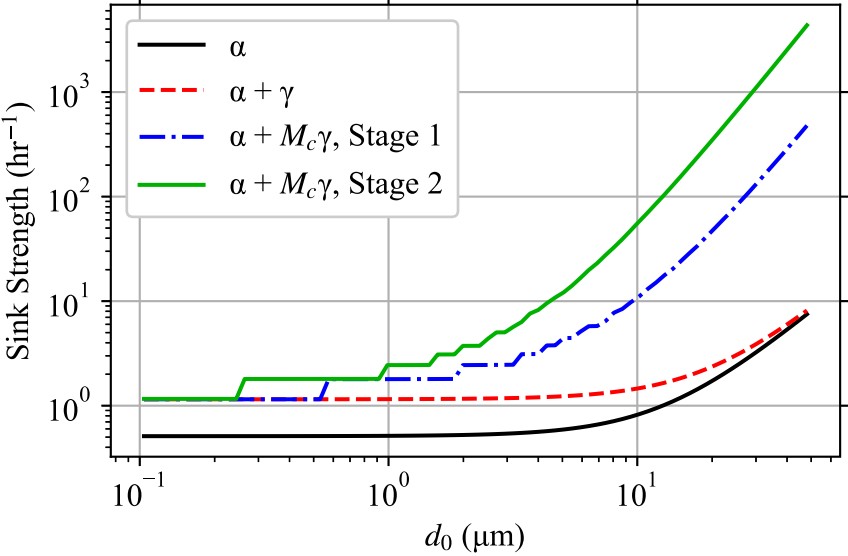

**Fig 4. Sink strength by bin.** The strength of the sink terms for each bin with 80 bins, which is $\alpha$ without inactivation, $\alpha + \gamma$ for $k = 1$, and $\alpha + M_c\,\gamma$ for $k = M_c$ (different values for Stage 1 and 2).

The pathogen concentrations as a function of $d_0$ and $k$ right after the beginning and at the end of each Stage are shown in S2 Fig. For large diameters, the concentrations at the beginning of each Stage are initially in a narrow band around the expected multiplicity in each diameter bin but by the end of each Stage the distributions have widened downward as inactivation fills in the lower multiplicities.

The results of choosing different numbers of bins (5, 20, and 80) is shown in S3 Fig. The difference in the concentration densities between 5 bins and 20 bins is substantial, but the difference between 20 and 80 is small. This means that in our example; for concentration densities, 20 bins is sufficient to capture the variation in $\alpha(d_0)$ and $\beta_k(d_0)$ with respect to diameter, but 5 is too few and 80 is a lot more effort for little gain. But for the $\mathfrak{R}_E$, the difference between the solutions for different number of bins is very small for the smaller $r = 2.45 \times 10^{-3}$, but more noticeable but still small for the larger $r = 5.39 \times 10^{-2}$.

## Discussion

### Effect of multiplicity on dose-response

We consider a few hypothetical examples to ellucidate the importance of multiplicity in the dose-response using the corrected exponential model in Eq (12). Another dose-response model could be chosen and the resulting values would differ, but the general pattern would be the same.

First, let's reconsider the example case but with all pathogen production forced to be mono-multiplicity. We set the new $\beta_{1,new} = \sum_{k=1}^{M_c} k\beta_k$ and all other $\beta_{k,new} = 0 \ \forall \ k \neq 1$ and then set $M_c = 1$ for all bins. This is equivalent to going to each bin, taking the total aerosol volume production, finding the expected number of pathogen copies in that volume, and redistributing the volume so that each pathogen is alone in an aerosol but not changing $d_0$ anywhere. Or put equivalently, making Eq (47) track pathogen copies instead of aerosols and ignoring multiplicity. To quantify the difference, we took a simplified version of the example where the second coughing infectious individual was removed, the $\rho_p$ of the first speaking infectious individual was adjusted, and we took the steady state case where $\vec{n}_0 = \vec{n}_\infty$ and calculated the constant $d\mu_{j,k}/dt$ for each susceptible individual. Then using the constant $d\mu_{j,k}/dt$ and an initial dose of zero, we found the time, $\tau_{50}$, required for $\mathfrak{R}_E$ to be 50% (note that the particular choice does not matter, the curve is identical for any chosen risk). This was calculated for the 80 diameter bins example to keep errors from finite bin width small, and a range of $r$ values up to the maximum value $r = 1$. Ignoring multiplicity causes $\tau_{50}$ to be underestimated (overestimation of risk). The underestimate of $\tau_{50}$ is shown in Fig 5.

The underestimation increases with increasing $\rho_p$ and $r$, and decreases when wearing a mask that is more efficient at filtering large aerosols than small aerosols. The largest aerosols have the greatest multiplicities, which means that a mask that filters them out better than small aerosols reduces the effect of ignoring multiplicity. As $\rho_p$ increases, the expected multiplicity range for each $d_0$ increases which makes ignoring multiplicity underestimate $\tau_{50}$ more. For the $r$ values considered here, $\rho_p \leq 10^9$ cm$^{-3}$ underestimates $\tau_{50}$ by at most 20% and $\rho_p \leq 10^8$ cm$^{-3}$ underestimates it by at most 12%. But for $\rho_p = 10^{11}$ cm$^{-3}$, the underestimation is up to 67%. To better understand these patterns, we need to consider two more hypothetical situations.

Let the average pathogen dose be $\langle \Delta \rangle = r^{-1}$ and all infectious aerosols have the exact same multiplicity $k$. Then, the $\mu$ for all other multiplicities is zero and $\mu_k = \langle \Delta \rangle / k$. Essentially, we are dividing the same number of pathogen copies among fewer and fewer aerosols as we increase the number of pathogen copies in each one. The mean infection risk for this constant average dose is shown on the left side of Fig 6 as a function of $k$ for four different $r$. As the multiplicity increases, the mean infection risk decreases even though the average dose is the same. For

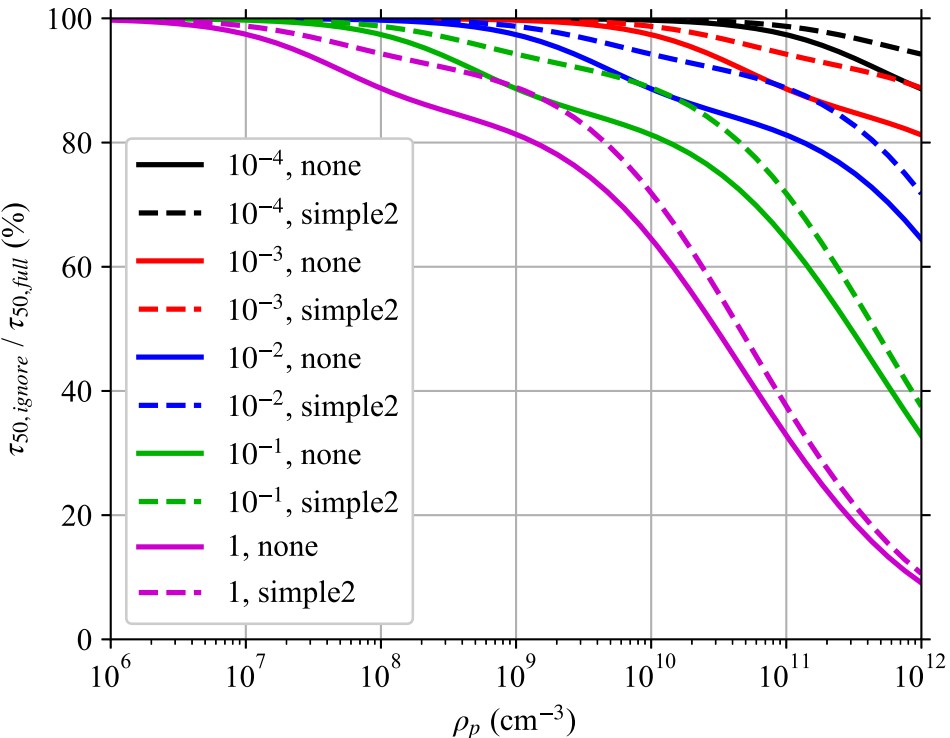

**Fig 5. Effect of ignoring multiplicity, full version.** Full version of Fig 1 with more $\rho_p$ and the effect of masks. Plot of the ratio of the time required to reach a 50% infection risk when multiplicity is ignored $\tau_{50,ignore}$ to when it is fully accounted for $\tau_{50,full}$ for different respiratory tract fluid pathogen concentrations $\rho_p$. We are considering the same situation as in the worked example, but at steady-state with just the speaking mask-less infectious individual and the risk to a susceptible individual whose exposure starts after steady state is reached. The ratio is shown for different combinations of mask on the susceptible individual (none and simple2) and for different $r$. The legend lists the $r$, mask combinations in the same order as the lines from top to bottom. We assumed a 100 nm diameter spherical pathogen and used 80 diameter bins and chose the $M_c$ (maximum multiplicity considered) heuristic threshold to be $T = 0.01$ (include 99% of pathogen production).

$k \ll r^{-1}$, the effect of multiplicity on $\mathfrak{R}_E$ is small. It starts to rapidly decrease near $k \sim r^{-1}$ and converges towards zero, because the number of pathogen copies in each aerosol is large enough that each aerosol has a high probability of causing infection by itself but the aerosols are decreasing in number faster than the risk can increase. The risk per aerosol can't exceed 100% no matter how many pathogen copies are in an aerosol.

Another way to see this is to consider another hypothetical. Let's consider the mean infection risk if all aerosols have multiplicity $k$ as we vary $r\langle\Delta\rangle$ for fixed $r$. This is shown on the right side of Fig 6 for $r = 10^{-2}$. For low $k \ll r^{-1}$, the infection risk curves are nearly identical. For $k \geq r^{-1}$, the infection risk decreases for increasing $k$.

Overall, this means that if the typical infectious aerosol multiplicity is on the order of or greater than $r^{-1}$, there can be a significant decrease in the infection probability for the same average dose. This has implications for large aerosols when the respiratory tract fluid pathogen concentration $\rho_{p,j}$ is large. Large aerosols where $\langle k \rangle \gtrsim r^{-1}$ will contribute less to the infection risk than would otherwise be expected from their resulting average pathogen dose $\langle\Delta_k\rangle$. While we must have $M_c > \langle k \rangle$, $M_c$ is usable as a proxy for which diameters the multiplicity causes a substantial correction to the dose-response. If we were to consider $r = 2.45 \times 10^{-3}$ as was done in the example, Fig 2 shows that this would be important for $d_0 > 15\ \mu m$ for a high viral concentration of $\rho_{p,j} = 10^{11}\ cm^{-3}$ and $d_0 > 30$ for the lower but still high viral concentration of

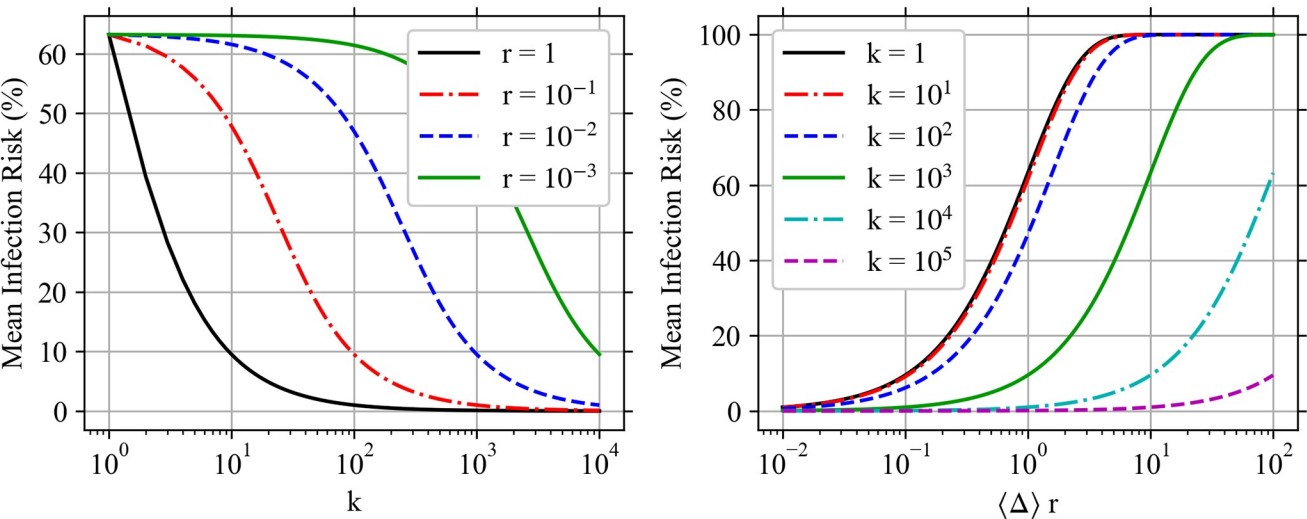

**Fig 6. Multiplicity's impact on infection risk.** Plots of mean infection risk ($\mathfrak{R}_E$) using the modified exponential dose-response model when all infectious aerosols have the same number of pathogen copies in them $k$. (Left) The infection risk as a function of $k$ for fixed average dose $\langle\Delta\rangle = r^{-1}$ for different single pathogen infection probabilities $r$. (Right) The infection risk as a function of the dose scaled by $r$ ($\langle\Delta\rangle r$) for different $k$ and the same fixed $r = 10^{-2}$ ($r^{-1} = 100$).

$\rho_{p,j} = 10^{10}$ cm$^{-3}$. If we were to consider $r = 5.39 \times 10^{-2}$ as was also done in the example, Fig 2 shows that this would be important for $d_0 > 5$ μm for a high viral concentration of $\rho_{p,j} = 10^{11}$ cm$^{-3}$ and $d_0 > 10$ for the lower but still high viral concentration of $\rho_{p,j} = 10^{10}$ cm$^{-3}$.

Going back to the risk overestimation from ignoring multiplicity in Fig 5, decreasing $r$ decreases the underestimation in $\tau_{50}$ because the ratio of the average multiplicity in the larger diameter bins to $r^{-1}$ is smaller. A mask that filters large aerosols better than small aerosols reduces the effect of ignoring multiplicity because larger aerosols have higher multiplicities.

## Filtering by the people

We introduced the sink terms $\alpha_{C,f}$ for filtering by the individuals in the environment as they inhale aerosols with many being absorbed by their mask or respiratory tract rather than being exhaled back out into the environment. To determine when this sink matters, we need to consider the total volume of air that is filtered, ignore the filtering efficiencies, and compare it to the ventilation. The volumetric rate of air filtration by the individuals normalized by the volume of the environment is

$$q_p(t) = \frac{1}{V}\left[\sum_{j=1}^{N_I}\lambda_{I,j}(t) + \sum_{j=1}^{N_S}\lambda_{S,j}(t) + \sum_{j=1}^{N_O}\lambda_{O,j}(t)\right] = \frac{\sigma_A}{\langle h\rangle}\langle\lambda_A\rangle_A, \tag{81}$$

where $\sigma_A$ is the horizontal area density of all individuals and $\langle h\rangle$ is the average height of the environment.

The mean adult breathing rates from sedentary/passive to high intensity activity ranges between 0.25 m$^3$ hr$^{-1}$ and 3.2 m$^3$ hr$^{-1}$ [14]. For sitting, it would be hard to get $\sigma_A$ to be more than 1 m$^{-2}$ but it would be possible while standing (some public events) though the well-mixed assumption would be breaking down in either case. For a typical room height of $\langle h\rangle = 4$ m, this density limit would yield max($q_p$)∈[0.063, 0.8] hr$^{-1}$. If the environment is poorly ventilated (total ventilation rate $q_v + q_o + q_r$ less than 1 hr$^{-1}$), this high people density would mean the filtering effect of the people would not be negligible compared to the ventilation. But with even

moderate ventilation, the contribution of $\alpha_{C,f}$ would be negligible unless all the ventilation is circulating ventilation ($q_o = q_r = 0$) with no filter or a very poor filter. For 1.5 and 2 m social distancing, the maximum $\sigma_A$ are 0.14 and 0.080 m$^{-2}$ respectively. For a typical room height of $\langle h \rangle$ = 4 m, this density limit would yield max($q_p$) $\in$ [0.005, 0.11] hr$^{-1}$ which would be negligible in almost all circumstances. For taller rooms, the contribution would be smaller if the total ventilation rate is held constant.

If the fraction of individuals who are infectious is held constant, then $N_I \sim \sigma_A$. Since $\beta_k \sim N_I$ and $\alpha_{C,f} \sim N_C$ but the non $\alpha_{C,f}$ terms of $\alpha$ stay constant, the source increases faster than the sinks meaning that $n_k$ increases and therefore $\mathfrak{R}$ increases. So, increasing $\sigma_A$ with everything else held constant increases the risk for the susceptible individuals. Thus, deliberately making $\alpha_{C,f}$ non-negligible is not a viable strategy to decrease risk. If the $\alpha_{C,f}$ dominate over the ventilation, the situation is actually quite hazardous from an infection transmission perspective. It is just that if one ignores the terms, one would overestimate the risk in such a crowded and poorly ventilated space.

## Effect of masks

The filtering effects of masks show up in the source $\beta_{I,k}$, the sinks $\alpha_{C,f}$, and the total dose over time $\mu_{j,k}$. Masks can substantially improve the total filtering efficiency of the people in $\alpha_{C,f}$ since aerosols have to pass through the mask twice, once on inhalation and again on exhalation at a larger diameter (many masks are better at filtering larger diameters than small diameters). But unless the ventilation is poor and there are a lot of people, this increase in $\alpha_{C,f}$ will have only a small effect on the total sink $\alpha$. Instead, the main contribution is to reducing $\beta_{I,k}$ and $\mu_{j,k}$ which are both linearly proportional to the mask survival efficiency, which can be seen in the example situation.

In the example during Stage 1, there is one infectious individual in the room who is not wearing a mask and the total pathogen concentration reaches about 40 m$^{-3}$ after 3 hr (Fig 3). During Stage 2, an addition infectious individual has entered the room. The second infectious individual's $\rho_p$ is 10 times greater than the first person's and they are breathing at 4 times the rate; which would mean 40 times the pathogen exhalation rate by itself. Additionally, they are coughing rather than speaking, with the resulting larger exhaled aerosol concentration density $\rho_j$ (top-right panel of Fig 3); which increases the number of exhaled pathogen copies further. But, they are wearing a mask which reduces the number of infectious aerosols that survive to reach the environment by a factor of 20–100 depending on the diameter. Due to this, the total pathogen concentration doesn't increase by a factor of over 40 but instead approximately triples, reaching approximately 140 m$^{-3}$.

The reduction in the average dose $\mu_{j,k}$ and therefore infection risk $\mathfrak{R}$ when susceptible individuals wear masks can also be seen in Fig 3. Even the simple1 mask gives some improvement, and the simple2 mask reduces the infection risk by over an order of magnitude.

Let's consider the case where all infectious individuals have the same mask survival efficiency and all susceptible individuals have the same mask survival efficiency. If the effects of masks on $\alpha$ is negligible ($\alpha_{C,f}$ is generally small compared to the other sinks) and $\beta_{r,k}$ is negligible; the combined effect of both infectious and susceptible individuals wearing masks on the dose is quadratic in the survival efficiencies, which has shown up in other Wells-Riley formulations in the past [7, 10]. Due to superposition of sources, $n_k \sim S_{I,m,out}$ since $\beta_{I,k} \sim S_{I,m,out}$. Then, $\mu_{j,k} \sim S_{S,m,in}\, n_k \sim S_{S,m,in}\, S_{I,m,out}$, which is a quadratic term. Now $\alpha_{C,f} \sim S_{C,m,in}\, S_{C,m,out}$ makes the effect stronger (usually only slightly stronger) than quadratic since it only serves to increase $\alpha$ and therefore decrease $n_k$ further. If everyone wears masks with the exact same survival efficiency $S$ for both inhalation and exhalation that is constant with respect to

$d_0$, then if exposure starts at steady state, $\mu_{j,k} \sim S n_{k,\infty} \sim S \beta_k / \alpha \sim S^2/(1 - cS^2)$ where $c \in [0, 1)$ is a constant that depends on the relative importance of the $\alpha_{C,f}$ in the total $\alpha$. In this form, it is easier to see how $\mu_{j,k}$ scales super-quadratically in the mask survival efficiency. If just the susceptible or just the infectious individuals wear masks, the reduction drops to being stronger than linear (direct contribution of the mask on reducing $\beta_{I,k}$ or reducing $\mu_{j,k}$ plus the effect on $\alpha_{C,f}$). If only non-susceptible non-infectious individuals wear masks, there is still a reduction in the dose but it is small since $\alpha_{O,f}$ is generally small compared to the other sinks, giving a sublinear reduction.

## Well-mixed limitation and corrections

The biggest limitation to the model presented here, like all Wells-Riley formulations, is the well-mixed environment assumption. In almost all indoor environments, the assumption breaks down to varying degrees—the infectious aerosol concentration densities at the locations of susceptible individuals and all sinks (except possibly inactivation) depend on their locations in the environment relative to the sources and the air flow. Social distancing helps with this assumption (reduces direct inhalation of undiluted exhaled puffs of aerosols from infectious individuals), but the assumption is still often dubious.

In situations where people, other sources, and localized sinks (or their outputs) are located close to each other; corrections to $n_k(d_0, t)$ must be applied at the location of the individual, other source, or sink. Here, we will qualitatively discuss what simple partial corrections that don't depend on the history of $n_k(d_0, t)$ would look like. For proximity to the output of filtering sinks, a multiplicative correction would need to be applied with a factor between the sink's filtering efficiency and one, inclusive, that depends on the location and the properties of the sink such as the flow rate. For proximity to the output of ventilation, the respective filtering efficiency is $E_v(w(d_0,t)d_0)$. For proximity to individuals, the respective filtering efficiency is $1 - S_{C,m,in,j} S_{C,r,j,k} S_{C,m,out,j,k}$. For proximity to sources, the correction would be to use a weighted average of $n_k(d_0, t)$ and the concentration of the air coming from the source/s with the weights depending on the location and the nature of the source flows and mixing, such as flow rates. For close proximity to ventilation coming from other rooms, this would mean a weighted average with $n_{r,k}(d_0, t)$ (if there is more than one room, it would be the concentration coming from the room/s whose air is not yet diluted at the location). For close proximity to infectious individuals, this would mean a weighted average with $n_{I,j,k}(d_0, t)[1 - E_{I,m,out,j}(d_0)]$. These partial corrections could be done for specific cases (e.g. susceptible individual 2 is 1 m directly in front of infectious individual 5) or in a statistical way if the pair correlation functions between individuals of each two categories (including in-category) as well as the equivalent correlation functions for relative angles of orientation by distance. More extensive corrections could depend on the history of $n_k(d_0, t)$ and would turn the system of ODEs into a system of Delay Differential Equations (DDEs) or Integro-Differential Equations (IDEs), which would most likely be much harder to solve. At some point, however, it could be easier to do a full fluid and aerosol dynamics treatment.

Any corrections developed for mono-multiplicity Wells-Riley formulations could either be used as is or could be adapted to the poly-multiplicity model presented in this manuscript. Full fluid dynamics simulations with infectious aerosols simulated as passive scalars or as discrete aerosols such as those done by Löhner *et al.* [40] are the common way to address this limitation entirely and can be used to develop corrections, which are considerably more difficult. Further investigation is needed to find simple approximate ways to generalize the Wells-Riley formulation presented in this manuscript for non-well-mixed environments that are easier than full fluid dynamics with suspended aerosols simulations.

## Other model limitations

Another limitation of the model presented here is that it assumes that all infectious aerosols have the same $\zeta$ and solute composition, and therefore the same $w(d_0, t)$. This is more easily circumvented in one case. If the solute concentration and composition is constant over time for each individual source (reasonable assumption over small time spans), the model can be solved for each source individually and then the resulting $n_k$ and $\mu_{k,j}$ summed over the individual solutions. This would also be the solution if $\zeta$ varies in different locations in the respiratory tract where infectious aerosols are produced for an infectious person. If $\zeta$ changes over time for the sources but the solute composition is constant, then one could generalize the model to additionally track $\zeta$ (or equivalently $d_D$) and initial diameter at production $d_0$ separately.

Another problem is the choice of diameter limits $d_0 \in [d_{m,k}, d_M]$ for each multiplicity. We have neglected the fact that the solute concentration is much greater for $d_0$ near the lower limit $d_{m,k}$ as pathogen copies are taking up a very large fraction of the volume and that surface effects may cause additional deviations in the number of pathogen copies in the aerosol from a Poisson distribution. Further work is needed to lift this limitation; though for small pathogens, the total fluid volume and therefore pathogen content in the smallest aerosols where this matters is much less than that of the larger aerosols (see top-right panel of Fig 3) meaning that the effect could be small for small pathogens.

The upper limit $d_M$ is the cutoff where aerosols are so large that they are more ballistic and either settle to the ground before evaporating to equilibrium or still settle too quickly to be mixed even after evaporating to their equilibrium diameter. Based on Xie *et al*. [25] and Chong *et al*. [21], we suggested a value $d_M$ = 50 μm. To look at it, we took the example case and re-calculated it for 23 equal log-width bins between 100 μm and 100 μm and considered the concentration densities and mean infection risks if the top 0, 2, and 4 bins were discarded, thereby setting decreasing $d_M$ to 100 μm, 54.8 μm, and 30.1 μm. The time step for the numerical solution had to be reduced to $5 \times 10^{-6}$ hr due to the increase in $M_c$ at the larger $d_M$. This is shown in Fig 7. Increasing $d_M$ increases the total pathogen concentration being tracked since a lot of exhaled respiratory tract fluid volume is contained in the large diameter aerosols, but the total number concentration does not increase much since these big aerosols are few in number. For the larger $r = 5.39 \times 10^{-2}$, the effect on $\mathfrak{R}_E$ is very small as $d_M$ is increased by a factor of approximately three. But for the smaller $r = 2.45 \times 10^{-3}$, there is a larger fractional difference in the mean infection risk but the additive difference is no more than 5% for the worst case (no mask). The masks as we have defined them in the example, are better at filtering large particles than small, so they attenuate the effect of increasing $d_M$ on $\mathfrak{R}_E$. More investigation is required on this upper diameter limit. Generalizing the model to track $d$ and $d_0$ and treating evaporation/growth explicitly over time would help alleviate this problem as the high settling rates and the slower evaporation of the largest aerosols could be treated explicitly.

## Conclusions

The number of pathogen copies in infectious aerosols must be taken into account if the number of pathogen copies in poly-multiplicity aerosols is not negligible compared to the number of pathogen copies in mono-multiplicity aerosols. We have generalized the Wells-Riley formulation and two common dose-response models (exponential and beta-Poisson) for poly-multiplicity aerosols and shown how to generalize other dose-response models. The generalized Wells-Riley formulation tracks infectious aerosols for each multiplicity individually rather than quanta as is traditional, which then can be put into the generalized dose-response model

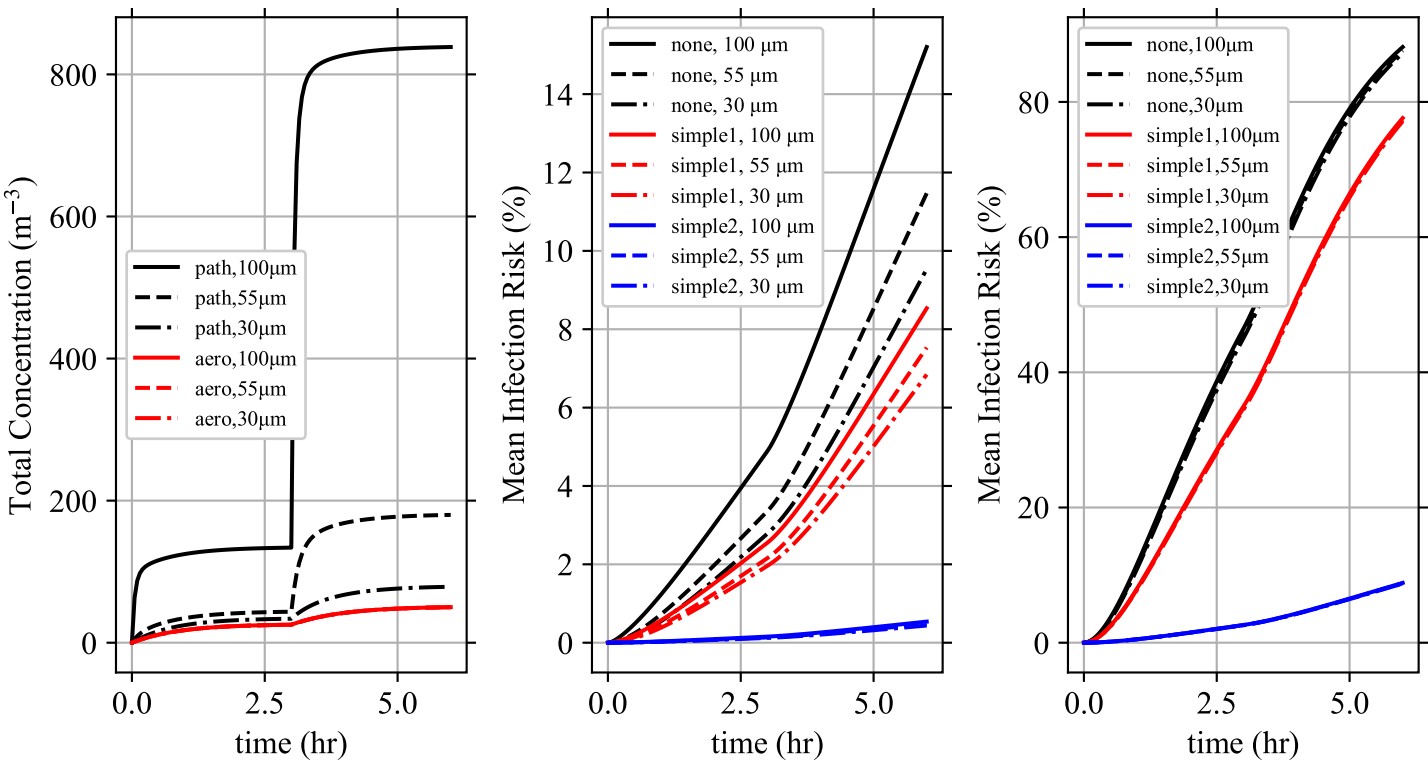

**Fig 7. Effect of upper diameter limit d$_M$.** The example situation was calculated for different values of the upper diameter limit $d_M$ (technically, calculated at the largest and then truncated down as needed). (Left) The total pathogen and infectious aerosol concentration densities over time for each $d_M$. Note that the differences in the total infectious aerosol concentration density are so small that the lines are right on top of each other. The mean infection risk for each combination of masks on a susceptible individual (none, simple1, simple2) for (Middle) $r = 2.45 \times 10^{-3}$ and (Right) $r = 5.39 \times 10^{-2}$.

of choice. The generalized Wells-Riley formulation results in a linear inhomogeneous coupled system of ODEs, one for each multiplicity, at each initial aerosol diameter at production $d_0$ (or bin of $d_0$). The general solution is presented; along with simplified versions for time independent sources, sinks, and humidity and splitting the diameter range into bins. The model is accompanied by an example case for for a poorly ventilated room with SARS-CoV-2, which is presented and solved. The example illustrates how the cutoff multiplicity $M_c$ is determined, the effects of bin size on the solution, and the effects of mask usage on the infection risk. Additional takeaways are

- Ignoring multiplicity causes the infection risk to be over-estimated, which is particularly signficant for high respiratory tract fluid pathogen concentrations and high single-pathogen infection probabilities (see Fig 5).

- The people in the environment filter the air by breathing, which increases the loss rate for infectious aerosols and is included in the model.

- Facemasks on everyone cause a stronger than quadratic reduction in the inhaled dose by susceptible individuals

In summary, we have developed a tractable generalization of the Wells-Riley model for the infection risk from any airborne disease in well mixed indoor environments applicable to both mono- and poly-multiplicity aerosols.

## Supporting information

**S1 Appendix. Model solution derivation.** Derivation of the general solution to Eq (47) as well as the constant in time coefficient special solution (both the explicit and recursive forms). (PDF)

**S2 Appendix. Checking analytical solution against numerical solution.** Checking the recursive analytical solution against solving the system of equations in Eq (47) numerically. (PDF)

**S3 Appendix. Binning diameter.** Shows how the model can be split into discrete diameter bins and each treated separately. (PDF)

**S4 Appendix. $M_c$ heuristic for infectious people derivation.** Derivation of the the individual infectious individual production heuristic for $M_c$ in Eq (79). (PDF)

**S5 Appendix. Numerical considerations.** Considerations for numerically evaluating the analytical model solution and solving the equations numerically; including how the number of terms scales with $M_c$ and the magnitude and precision requirements to avoid numerical overflow and losing accuracy. (PDF)

**S1 Fig. Filtering efficiencies of simple1 and simple2 masks from the example situation.** The filtering efficiencies of the simple1 and simple2 masks from the example, whose functional forms are given by Eq (81), as a function of the diameter. (PDF)

**S2 Fig. Pathogen concentration by k and diameter for the example situation.** The pathogen concentration in the room as a function of $d_0$ and $k$, denoted by color, at four different times (listed in the title of each panel) in the example situation. They are (Top-Left) right after the beginning of Stage 1, (Top-Right) at the end of Stage 1, (Bottom-Left) right after the beginning of Stage 2, and (Bottom-Right) at the end of Stage 2. All four panels share the same colorbar, which is in the bottom-right panel. (PNG)

**S3 Fig. Comparing different numbers of bins in the model solution's for the example situation.** Version of Fig 3, but comparing the model solution for the example situation for 5, 20, and 80 bins. (Top-Left) The total pathogen and infectious aerosol concentrations over time for each number of diameter bins used to solve the model. (Top-Right) The infectious aerosol concentration densities as a function of $d_0$ at $t = 6$ hr for each number of bins. (Bottom-Left, Bottom-Right) The mean infection risk $\mathfrak{R}_E$ for the susceptible individuals based on the mask they are wearing (none, simple1, or simple2) for each number of bins using (Bottom-Left) $r = 2.45 \times 10^{-3}$ and (Bottom-Right) $r = 5.39 \times 10^{-2}$. (PDF)

## Acknowledgments

We would like to thank Oliver Schlenczek for important discussions early in the development of the model, Jan Moláček for comments and discussion during editing, and Hani Kaba and Simone Scheithauer for useful references and discussing those references.

## Author Contributions

**Conceptualization:** Freja Nordsiek, Eberhard Bodenschatz, Gholamhossein Bagheri.

**Data curation:** Freja Nordsiek, Gholamhossein Bagheri.

**Formal analysis:** Freja Nordsiek, Eberhard Bodenschatz, Gholamhossein Bagheri.

**Funding acquisition:** Eberhard Bodenschatz.

**Investigation:** Freja Nordsiek, Eberhard Bodenschatz, Gholamhossein Bagheri.

**Methodology:** Freja Nordsiek, Eberhard Bodenschatz, Gholamhossein Bagheri.

**Resources:** Eberhard Bodenschatz.

**Software:** Freja Nordsiek.

**Validation:** Freja Nordsiek, Gholamhossein Bagheri.

**Visualization:** Freja Nordsiek.

**Writing – original draft:** Freja Nordsiek.

**Writing – review & editing:** Freja Nordsiek, Eberhard Bodenschatz, Gholamhossein Bagheri.

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
