## [Decision Letter · Decision Letter 0]

18 Jan 2021

PONE-D-20-37387

Risk assessment for airborne disease transmission by poly-pathogen aerosols

PLOS ONE

Dear Dr. Bodenschatz,

Thank you for submitting your manuscript to PLOS ONE. We have received two reviewers' reports, both of which contain minor comments. We invite you to submit a revised version of the manuscript that addresses the points raised during the review process, after which, an immediate acceptance is possible.

We look forward to receiving your revised manuscript.

Kind regards,

Ivan Kryven

Academic Editor

PLOS ONE

Journal Requirements:

2.Thank you for stating the following financial disclosure:

Reviewers' comments:

Reviewer's Responses to Questions

**Comments to the Author**

1. Is the manuscript technically sound, and do the data support the conclusions?

Reviewer #1: Yes

Reviewer #2: Yes

2. Has the statistical analysis been performed appropriately and rigorously? 

Reviewer #1: I Don't Know

Reviewer #2: N/A

3. Have the authors made all data underlying the findings in their manuscript fully available?

Reviewer #1: Yes

Reviewer #2: Yes

4. Is the manuscript presented in an intelligible fashion and written in standard English?

Reviewer #1: Yes

Reviewer #2: Yes

5. Review Comments to the Author

Reviewer #1: This is an interesting paper that proposes a revised model for the risk assessment of airborne virus transmission by poly-pathogen aerosols. The paper is generally well written, the analysis seems sound, though note I am not an expert in the statistical description presented in this paper.

Overall, I can recommend publication of this paper, as it introduces an interesting perspective on a very timely and important problem. There are, however, some points I recommend the Authors to consider prior to acceptance:

- The level of the English language is overall good, but there are repeated typos all over the manuscript. Please make sure you amend those typos.

- In the Introduction, it is stated that SARS-CoV-2 is an airborne virus. Actually, there is still much debate on that, i.e, to what extent SARS-CoV-2 can be transmitted indirectly at long range. Please add a reference to support this claim.

- The paper deals mostly with the statistical analysis of the phenomenon, in my view overlooking the equally important fluid dynamics behind the generation of the expiratory cloud and aerosolisation. I recommend the Authors to at least mention these important aspects. There is an increasing number of publications available in the literature, see for example:

1 L. Bourouiba, “Turbulent gas clouds and respiratory pathogen emissions. Potential implications for reducing transmission of COVID-19,” JAMA Insights 323, 1837 (2020).

2 R. Mittal, R. Ni, and J.-H. Seo, “The flow physics of COVID-19,” J. Fluid Mech. 894, F2 (2020).

3. E. Renzi, A. Clarke "Life of a droplet: Buoyant vortex dynamics drives the fate of micro-particle expiratory ejecta," Phys. Fluids 32, 123301 (2020)

Reviewer #2: The authors have presented an analytical approach to risk-assessment of respiratory droplet laden diseases. The manuscript's main finding is that mono-pathogen approximation largely overpredicts the risk, while the poly pathogen model is more realistic. The model is well-defined and includes several thermo-physical (droplet transport, evaporation, settling, etc.) and behavioral aspects (effect of masks, inhalation, exhalation, etc.).

The study is appropriate for the journal and contains exciting results, which will help plan for strategies to fight airborne diseases. The reviewer has a few comments. Once these are addressed, the manuscript can be recommended for publication.

• While the introduction discusses some essential research in the field, the reviewer feels the authors should also discuss a few recent developments in the context. Recently the transport and fluid mechanistic aspects of the respiratory droplets have received great attention. For example, the details of the transport, settling, and evaporation of the droplets [1-6]; the effect of speech and plosive sounds on respiratory droplets transport and risk of transmission [7-8]; physics disease transport modeling from droplet dynamics [9-10] have been explored in great details. A summary of these studies could benefit the readers of the current paper since they can add more granularity to the model presented.

• The Wells-Riley model's primary assumption is that the pathogens (via droplets) are homogeneously distributed in space. However, these pathogens' concentration is expected to be higher close to an infected individual and lower close to a healthy individual. Does the presented model account for such inhomogeneity? If not, can this be included in the as an additional effect?

[1]. R. Mittal, R. Ni, J.-H. Seo, J. Fluid Mech., 2020, 894, F2.

[2]. T. Dbouk, D. Drikakis, Phys. Fluids, 2020, 32, 053310.

[3]. L. Bourouiba, Jama, 2020, 323(18), 1837

[4]. H. Li, F.Y. Leong, G. Xu, Z. Ge, C. W. Kang, K. H. Lim, Phys. Fluids, 2020, 32(11), 113301.

[5]. S. Chaudhuri, S. Basu, P. Kabi, V. R. Unni, A. Saha, Phys. Fluids, 2020, 32(6), 063309.

[6]. S. Balachandar, S. Zaleski, A. Soldati, G. Ahmadi, L. Bourouiba, Int. J. Multiphase Flow, 2020, 132, 103439

[7]. M. Abkarian, S. Mendez, N. Xue, F. Yang, H. A. Stone, Proc. National Acad, Sc., 2020, 117(41), 25237-25245.

[8]. F. Yang, A. A. Pahlavan, S. Mendez, M. Abkarian, H. A. Stone, Phy. Rev. Fluids, 2020, 5, 122501.

[9]. S. Chaudhuri, S. Basu, A. Saha, Phys. Fluids, 2020, 32(12), 123306.

[10]. R. Mittal, C. Meneveau, W. Wu, Phys. Fluids, 2020, 32, 101903.

6. PLOS authors have the option to publish the peer review history of their article (what does this mean?). If published, this will include your full peer review and any attached files.

Reviewer #1: No

Reviewer #2: No

---

## [Author Response · Author response to Decision Letter 0]

17 Feb 2021

The comments can be found in a file that is uploaded

---

## [Editor Report · Decision Letter 1]

18 Feb 2021

Risk assessment for airborne disease transmission by poly-pathogen aerosols

PONE-D-20-37387R1

Dear Dr. Bodenschatz,

Thank you for incorporating reviewers feedback and additional changes to the manuscript. We are pleased to inform you that your manuscript has been judged scientifically suitable for publication and will be formally accepted for publication once it meets all outstanding technical requirements.

Kind regards,

Ivan Kryven

Academic Editor

PLOS ONE
---

## [Editor Report · Acceptance letter]

5 Mar 2021

PONE-D-20-37387R1 

Risk assessment for airborne disease transmission by poly-pathogen aerosols 

Dear Dr. Bodenschatz:

I'm pleased to inform you that your manuscript has been deemed suitable for publication in PLOS ONE. Congratulations! Your manuscript is now with our production department. 

Kind regards, 

on behalf of

Dr. Ivan Kryven 

Academic Editor

PLOS ONE